# Internally generated time in the rodent hippocampus is logarithmically compressed

Rui Cao[1]*[†], John H Bladon[2†], Stephen J Charczynski[1], Michael E Hasselmo[1], Marc W Howard[1]

[1]Department of Psychological and Brain Sciences, Boston University, Boston, United States; [2]Department of Psychology, Brandeis University, Waltham, United States

**Abstract** The Weber-Fechner law proposes that our perceived sensory input increases with physical input on a logarithmic scale. Hippocampal 'time cells' carry a record of recent experience by firing sequentially during a circumscribed period of time after a triggering stimulus. Different cells have 'time fields' at different delays up to at least tens of seconds. Past studies suggest that time cells represent a compressed timeline by demonstrating that fewer time cells fire late in the delay and their time fields are wider. This paper asks whether the compression of time cells obeys the Weber-Fechner Law. Time cells were studied with a hierarchical Bayesian model that simultaneously accounts for the firing pattern at the trial level, cell level, and population level. This procedure allows separate estimates of the within-trial receptive field width and the across-trial variability. After isolating across-trial variability, time field width increased linearly with delay. Further, the time cell population was distributed evenly along a logarithmic time axis. These findings provide strong quantitative evidence that the neural temporal representation in rodent hippocampus is logarithmically compressed and obeys a neural Weber-Fechner Law.

**\*For correspondence:**
caorui.beilia@gmail.com

[†]These authors contributed equally to this work

**Competing interest:** The authors declare that no competing interests exist.

## Editor's evaluation

This is a rigorous evaluation of whether the compression of time cells in the hippocampus follows the Weber-Fechner Law, using a hierarchical Bayesian model that simultaneously accounts for the firing pattern at the trial, cell, and population levels. The two key results are that the time field width increases linearly with delay, even after taking into account the across trial response variability, and that the time cell population is distributed evenly on a logarithmic time scale.

## Introduction

The Weber-Fechner law states that there is a logarithmic relationship between the perceived intensity of a sensory stimulus and the actual intensity (*Fechner, 1860*). The Weber-Fechner law provides a good description of a wide range of phenomena in psychology including perceptual phenomena (*Moore, 2012*), the non-verbal numerosity system (*Feigenson et al., 2004*; *Gallistel and Gelman, 2000*), and psychological studies of time perception (*Rakitin et al., 1998*; *Gibbon et al., 1984*). It has been proposed that these psychological phenomena reflect logarithmically compressed neural representations of time, space, and number (*Gallistel, 1989*; *Dehaene and Brannon, 2011*). It has long been known that the representation of visual space in the retina and early visual cortical regions do not map visual space veridically. Rather, the representation of visual space is logarithmically compressed as a function of distance from the fovea (*Schwartz, 1980*; *Van Essen et al., 1984*; *Glezer, 1965*). Moreover, neurons in monkey parietal cortex recorded during numerosity judgements

**Figure 1.** Temporal representation and the Weber-Fechner law. (**a**) A compressed timeline of the past. The horizontal line A B C … represents objective reality as the experimenter presents a series of stimuli at regularly spaced intervals. At each moment, memory carries a subjective representation of what happened and when (diagonal line). The time of more recent events are represented more clearly while distant events are less distinguishable from each other, resulting in a compression of the temporal memory. (**b**) A schematic illustration of a neural implementation of a logarithmically-compressed timeline. Each curve corresponds to a time field. When plotted as a function of $\log$ time (bottom), the receptive fields appear evenly distributed. When plotted as a function of linear time (top), the time-fields stretch out and become less numerous as time passes. (**c**) Heat map of hypothetical time cells firing pattern if time is represented logarithmically. (**insert**) Time-field widths are linearly correlated with time-field peaks. Each row represents the firing pattern of a time cell, sorted by the peak of its time-fields. Dark blue indicates low activity while light yellow indicates high activity. Because time-fields follow logarithmic compression, the sorted peaks form a specific curved line where more time cells fire at the beginning of the delay. In the insert, the time-field widths are plotted as a function of their peaks. Logarithmic compression predicts a linear relationship.

have receptive fields organized as a function of the logarithm of numerosity (*Nieder and Miller, 2003*). Similar findings are observed in human fMRI (*Cantlon et al., 2006*), suggesting a common system across species (*Nieder and Dehaene, 2009*; *Brannon, 2006*). A recent paper from a cerebellar slice preparation argues that unipolar brush cells form a logarithmically-compressed set of temporal basis functions (*Guo et al., 2021*). This paper asks the question whether hippocampal time cells form a logarithmic code for the time of past events, consistent with a Weber-Fechner representation of past time.

## Time cells in the hippocampus

So-called "time cells" in the hippocampus fire sequentially during unfilled delay intervals (*Pastalkova et al., 2008*; *MacDonald et al., 2011*). Because the sequence is reliable across trials, one can reconstruct the time at which the delay began (e.g. *Mau et al., 2018*) and, in many cases, the identity of a stimulus that initiated the delay (*Taxidis et al., 2020*; *Cruzado et al., 2020*) such that at the population level hippocampal time cells construct a record of what happened when (*Figure 1a*). It is well-established that time cells do not provide a veridical record of the past; rather the accuracy of the representation of time within the delay decreases with the passage of time (*Kraus et al., 2013*; *Mau et al., 2018*; *Taxidis et al., 2020*). However, unlike the visual system, which clearly shows logarithmic compression, the quantitative form of the compression for time expressed in hippocampal time cells is not known.

## Quantifying logarithmic compression

Logarithmic compression leads to two quantitative relationships (*Figure 1b and c*) that have been empirically demonstrated in the visual system. First, the size of receptive fields grows linearly with distance from the fovea (*Dumoulin and Wandell, 2008*; *Gattass et al., 1981*). Second, the number of cells with receptive fields in a region of visual space goes down as the inverse of distance from the fovea (*Daniel and Whitteridge, 1961*; *Hubel and Wiesel, 1974*; *Van Essen et al., 1984*). To see why logarithmic compression leads to these two properties, let us suppose that we have a set of time cells that fire in sequence following some event at $t = 0$. Let's order the time cells according to the time at which they peak firing and call the time at which the $nth$ time cell fires maximally as $t_n$. We say the time cells form a logarithmically compressed representation of time if

$$n = \log_b \left( \frac{t_n}{t_0} \right) \tag{1}$$

where $t_0$ is the peak time of the earliest time cell and $b > 1$ is the base of the logarithm. *Figure 1b* gives an example where $t_0 = 1$ and the $b = 2$ giving time field peaks at $1, 2, 4, 8 \ldots$.

Both of the quantitative properties of logarithmic compression follow naturally from *Equation 1*. *Equation 1* implies $t_n = t_0 b^n$ and $t_{n+1} = t_0 b^{n+1}$ for all choices of $n$. Notice that the ratio between adjacent time cell peaks is constant (i.e. $b$) for all $n$. Let us define $\Delta_n$ as the distance between $t_n$ and $t_{n+1}$. So if the time cell centers are logarithmically compressed (*Equation 1*), this implies that the spacing between adjacent time fields goes up proportional to the center of the time field for all $n$:

$$\Delta_n \equiv t_{n+1} - t_n = (b-1)t_n \tag{2}$$

The first quantitative relationship describes the width of temporal receptive fields as a function of peak time. Let us write $\sigma_n$ to describe the width of the temporal receptive field of receptor $n$ that peaks at $t_n$. We have seen that logarithmic compression, *Equation 1*, requires that $\Delta_n$, the delay between the peaks of adjacent time cells, goes up linearly with peak time, *Equation 2*. Consider the overlap between adjacent temporal receptive fields. If $\sigma_n$ grew more slowly than $\Delta_n$, this would mean that the overlap of the temporal receptive fields would decrease with $n$. In this case the set of receptors would represent longer times with progressively less resolution. Conversely, if $\sigma_n$ grew more rapidly than $\Delta_n$, the overlap of the receptive fields of adjacent time cells would increase with $n$. Both of these situations are suboptimal. In contrast if $\sigma_n$ grows like $\Delta_n$, the population will cover all parts of the timeline with similar resolution. Because $\Delta_n$ goes up linearly with $t_n$, these considerations lead to the property of a logarithmic timeline that

$$\sigma_n \propto t_n \tag{3}$$

*Equation 2* also specifies the relative proportion of time cells that peak at a particular time. Let us assume that there is a very large number of time cells covering a finite interval of time. Let us consider a small neighborhood around a particular value $\tau$ within this interval. If the spacing between time field peaks in the region around $\tau$ is small, we would find a lot of time cell peaks. Conversely, if the spacing between time field peaks in the region around $\tau$ is large, we would find fewer time cells peaking in the neighborhood around $\tau$. *Equation 2* says that the spacing between adjacent time cells goes up like $\tau$. It stands to reason that the number of time cells we find in a neighborhood around $\tau$ should go down like $\tau^{-1}$. Suppose we pick a time cell at random from the population tiling an interval. The probability of finding that that time cell's peak time is $t$ should obey

$$p(t) \propto t^{-1} \tag{4}$$

where the constant of proportionality depends on the range of times across the population.

## Are time cells *logarithmically* compressed?

The goal of this paper is to rigorously test the hypothesis that time cells form a logarithmically-compressed representation of past time by evaluating these two quantitative predictions (*Equation 3 and 4*). This requires more subtle analyses than in previous time cell studies. For instance, many prior hippocampal time cell studies simply estimate the receptive field width by averaging over trials. It is conceivable that the increase in receptive field width is simply an averaging artifact due to variability in receptive field location across trials (*Figure 2b*).

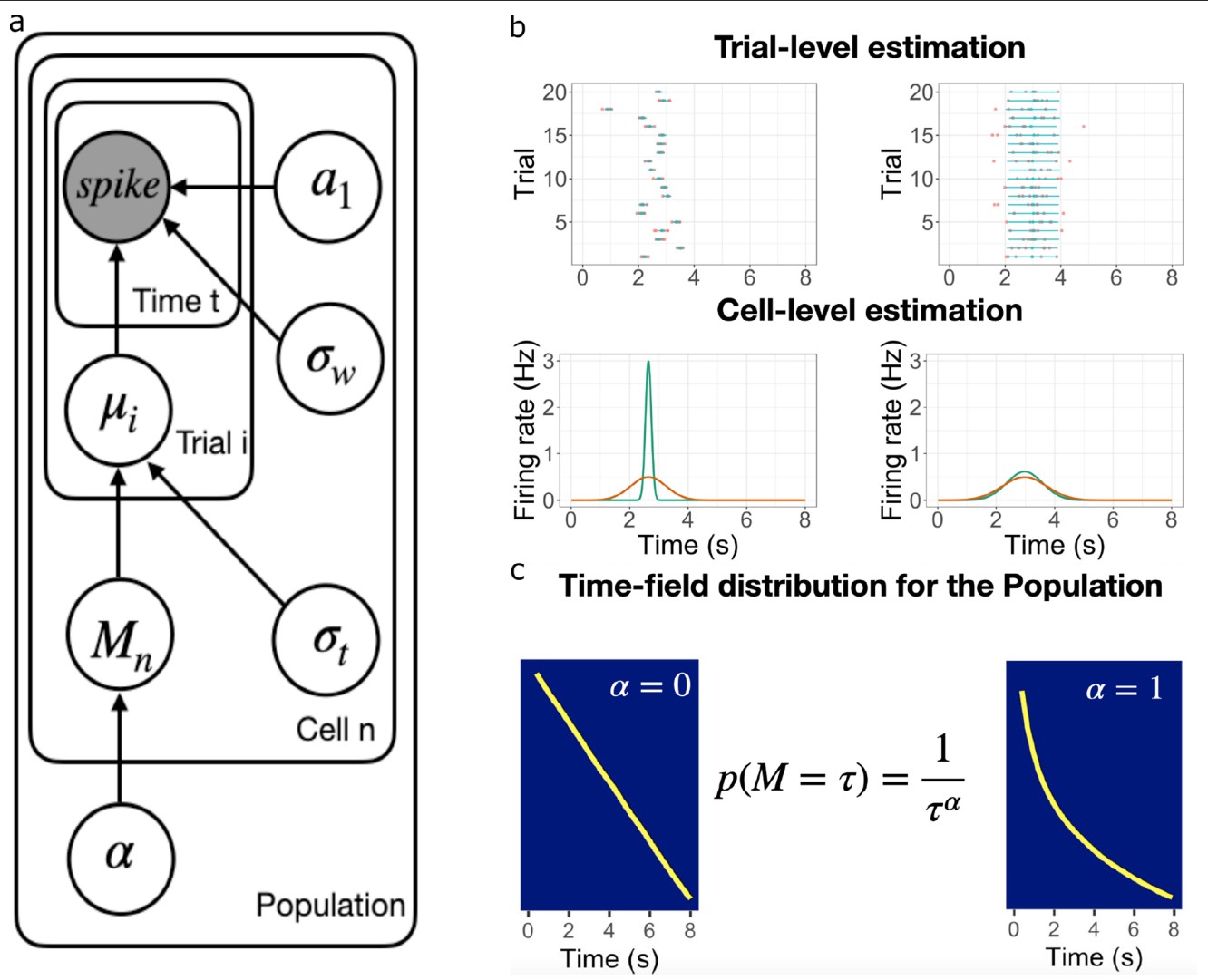

**Figure 2.** Schematic illustration of the best fitting hierarchical model. (**a**) Graphic diagram of the model. Each node represents a variable in the model; the filled node represents the observed spike-train data and open nodes represent latent variables. Arrows represent relationships between variables and plates indicate whether the variable is estimated at the trial level, cell level, or population level. (**b-c**). Schematic illustration of model fitting for a group of simulated time cells. The model accounts for the firing pattern at individual trials (top rows in **b**), individual cells (bottom rows in **b**) and the population level c. (**b**) Two simulated cells with high (left) vs low (right) variability in time field location across trials. The pink dots indicate the spike train and the blue lines indicate the estimated time fields for each trial. At the cell level, the red lines indicate the average firing patterns across trials and the green lines indicate the estimated time fields for the cell from the model. For both the left and right cells, the overall firing pattern is similar and would result in similar time fields averaged over trials. By contrast, fitting at the individual trial level allows the model to estimate the within-trial time field. (**c**) Effect of $\alpha$ on time field distributions. The prior for finding a time field mean $M$ goes like $M^{-\alpha}$. The yellow line plots the location of the time field peaks, sorted from early to late, for two specific values of $\alpha$. Left: $\alpha = 0$ gives a uniform distribution such that all points in the past are equally represented by time cells. Right: $\alpha = 1$ resulting in logarithmic compression of the time axis.

We recorded from dorsal CA1 during a memory experiment in which rats had to remember the identity of an object across an eight-second delay (*Figure 3*). This task is known to produce robust time cells, which we identified using standard methods. We estimated the distribution of time field parameters simultaneously at the levels of individual trials, individual units (across trials), and the population (across units) using a hierarchical Bayesian model (*Figure 2*). In hierarchical Bayesian modeling, each level of the hierarchy serves as a prior for parameter estimates at lower levels; parameters of the model at all levels are adjusted to maximize the likelihood of the entire observed dataset (*Lee, 2011*). This approach allows us to separately estimate variability in receptive field centers *across* trials and the

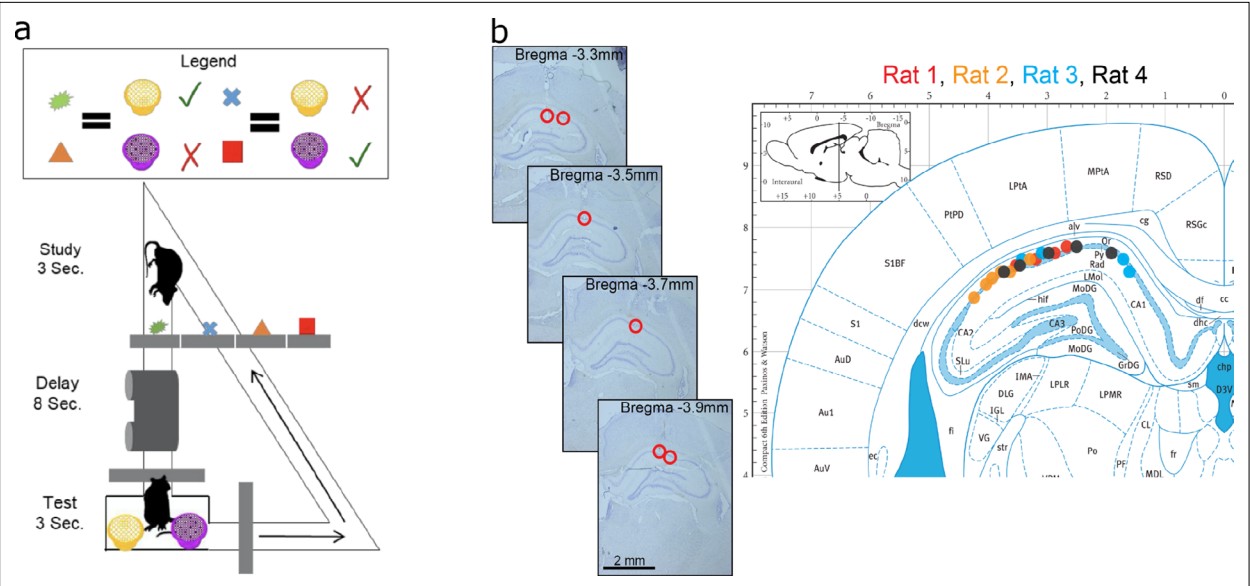

**Figure 3.** Data acquisition. (**a**) Task paradigm. Rats performed a delayed-matching task wherein each of the four study objects indicated which of the two test objects contained hidden reward. (**b**) Recording sites. The left panel plots example histology sections for animal 1. The red circle indicates the final recording location. The right panel plots the estimated tetrode locations for each rat (coded by different color dots) mapped onto the Paxinos and Watson atlas (1986).

The online version of this article includes the following figure supplement(s) for figure 3:

**Figure supplement 1.** Within trial behavioral variability decreases with delay.

---

width of receptive fields *within* trials and thus test both of the quantative predictions (*Equations 3; 4*) that follow from logarithmic compression.

## Results

Each trial started with rats exploring an object for 3 s, followed by an 8 s delay during which rats ran on a treadmill (*Figure 3a*). After the delay, rats were presented with two terra cotta pots with unique odors and digging media. The identity of the object the animal explored at the beginning of the delay specified which of the pots contained reward. Rats learned the memory task well with accuracy between 70 and 80%. There was very little variability in positions or other measurable changes in behavior after the first two seconds of the delay (*Figure 3—figure supplement 1*, detailed analysis in Supplemental section S1).

Among the total of 625 pyramidal CA1 units we recorded from the four animals, 159 units (30/87 for rat 1; 39/134 for rat 2; 32/111 for rat 3; 58/293 for rat 4) were classified as time cells using previous methods (*Tiganj et al., 2018*). The proportion of time cells is similar to previous studies using similar methods (e.g. *Salz et al., 2016*). Fourteen time cells whose peak times were considered too close to the boundaries of the delay interval (see Materials and methods for details) were removed. Because hierarchical Bayesian models estimate across-trial variability separately from within-trial variability, another 14 units were removed for low trial-count (<20 trials). The detailed procedures used to select time cells for further model estimation can be found in the Materials and methods section. A total of 131 robust time cells were considered further for the hierarchical Bayesian analyses.

Below we first briefly describe the hierarchical Bayesian models considered to describe the firing patterns of time cells across trial, cell and population levels ('Estimating time fields …'). Then we report the model results of these models in two sections. The section titled 'The representation of time varies …' focuses on variability in time fields across trials. The following section ('Hippocampal time cells form a …') focuses on the properties at the population level and provides direct tests of the hypothesis that the hippocampal time code is logarithmically compressed.

**Table 1.** WAIC of the Main Model and alternative models.

|  | Time field | Trial variability | Population | WAIC |
|---|---|---|---|---|
| Main Model | Gaussian | Location shift | Power-law | 1.4e+18 |
| Log-normal Time Field Model | Log-normal | - | - | 4.8e+19 |
| Alternative Trial Vary Model | - | Width vary | - | 6.3e+27 |
| Exponential Compression Model | - | - | Exponential | 5.2e+26 |
| Weibull Compression Model | - | - | Weibull | 1.7e+25 |

Note: - indicates the same assumption as the Main Model.

## Estimating time fields with hierarchical Bayesian method

For expository purposes, we first sketch out the model that best describes the time fields across all levels (*Figure 2*). Note that we tested alternative assumptions at every level of the hierarchy (see *Table 1* for reference) and evaluated the models via Widely Applicable Information Criterion (*Watanabe and Opper, 2010*). We refer to the best-fitting model as the 'Main Model'. The detailed fitting procedures for all models can be found in the Materials and methods section.

Unlike previous time cell studies that required a single time field across all trials, the models we tested allow for separate time field estimation for each trial. Given $m$ spike times $t_1...t_m$ on trial $i$ for a time cell, the model assumes the observed spikes are a mixture of two distributions. A portion $a_1$ of spikes is sampled from a Gaussian-shaped time field centered at $\mu_i$ with standard deviation $\sigma_w$. The remaining proportion, $1 - a_1$ is sampled from a uniform distribution over the length of the entire delay $l$. Therefore the probability that one of the spikes happened is at time $t_j$ is

$$p(t_j|a_1, \mu_i, \sigma_w) = a_1\mathcal{N}(t_j|\mu_i, \sigma_w^2) + \frac{(1-a_1)}{l} \tag{5}$$

An alternative model assumes that the time field follows a log-normal distribution ("Log-normal Time Field Model", *Table 1*).

Note that this approach assumes that each spike is independently sampled from the mixture of distributions rather than explicitly modeling a changing firing rate. This approach ignores dependencies in spike times in favor of a simplified time field estimation that feeds into the next levels of the hierarchical model. This simplified trial-level assumption is adequate for characterizing the distribution of time field parameters. Given the independent sampling assumption, the likelihood of producing the observed set of spikes is:

$$\mathcal{L}_i = \prod_{j=1}^m p(t_j|a_1, \mu_i, \sigma_w^2) \tag{6}$$

To account for variability across trial, $\mu_i$ is assumed to be distributed normally across trials:

$$\mu_i \sim \mathcal{N}(M, \sigma_t^2) \tag{7}$$

Therefore, the mean $M$ of this distribution estimates each time cell's average peak time. The standard deviation of this distribution, $\sigma_t$, estimates the variability of that cell's time field across trials. By separating the trial variability $\sigma_t$ from the estimate of the time field width within trial $\sigma_w$, we are able to investigate the relationship between $\sigma_w$ and time field peak $M$ as logarithmic compression predicts that time field width within a trial goes up linearly with time field center. We also considered the possibility that the width of time fields varies across trials rather than the mean ('Alternative Trial Vary Model' in *Table 1*).

Finally, the time field locations $M$ across the CA1 population was assumed to follow a power-law distribution:

$$P(M = \tau) \propto \tau^{-\alpha} \tag{8}$$

The Main Model returns the posterior distribution of $\alpha$ given the data. That means the Main Model is capable of observing logarithmic compression—a tight posterior distribution around $\alpha = 1$—but also other hypotheses. For instance, $\alpha = 0$ corresponds to a uniform distribution of time field locations

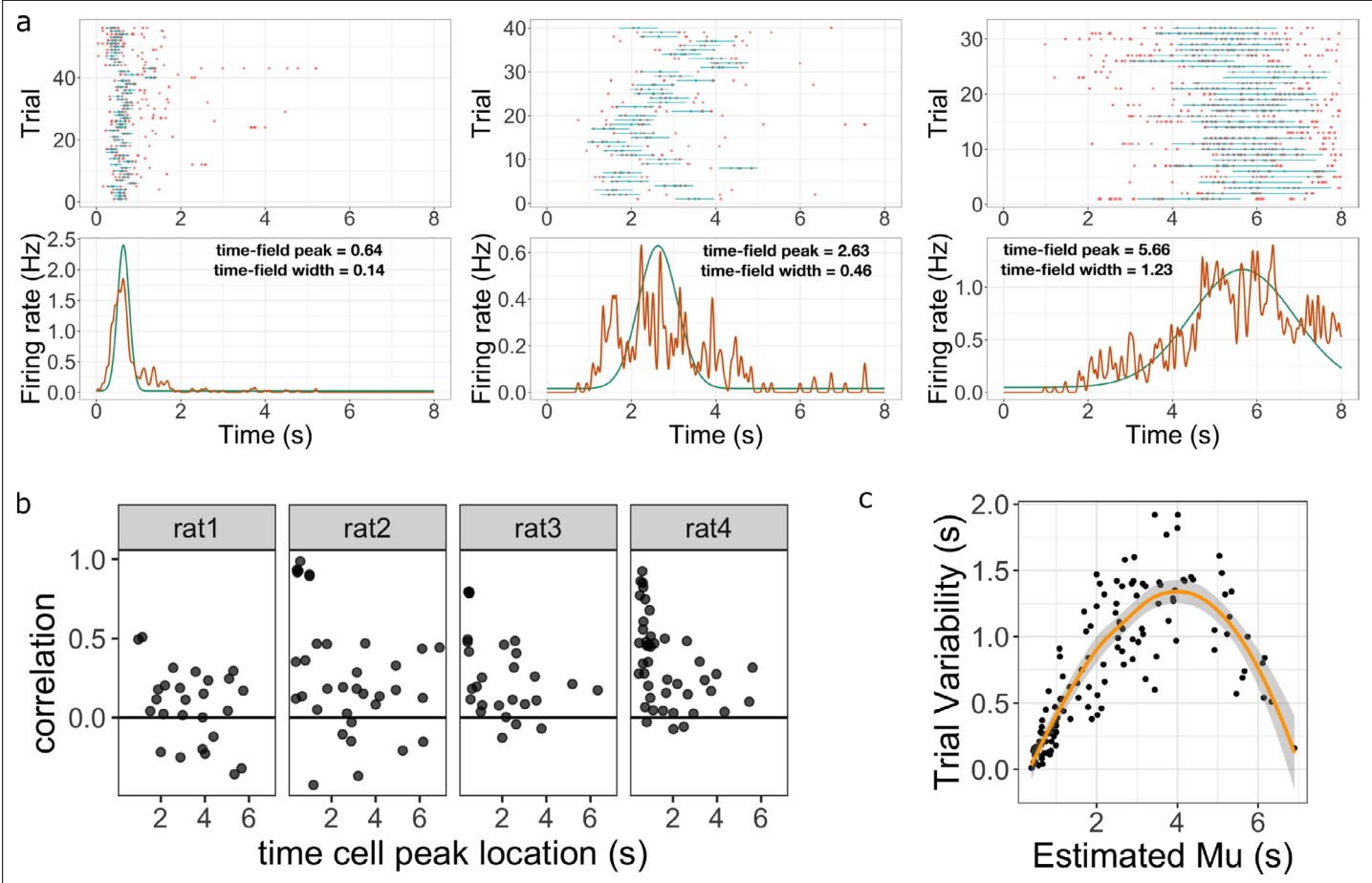

**Figure 4.** Across-trial variability measured by the Main Model. (**a**) Three representative time cells with different temporal receptive fields. Rasters show estimated time fields for each trial as a blue line. In the bottom plots, the red line represents the peristimulus-time histogram averaged over trials. The green line indicates the estimated time field with within-trial time field width ($\sigma_w$, shown above). Across-trial variability in the location of the time field is estimated separately. (**b**) Trial-to-trial variability in time field location is correlated with the timing of events early in the trial. Several events happen at the beginning of each trial; the door opens to allow access to the treadmill, the animal steps onto the treadmill and then breaks a beam at the middle of the treadmill. The last event is used as time zero in this study. The y-axis in these graphs shows the Pearson correlation between time field peak location and the estimate of the time rats stepped on the treadmill on each trial. Each dot represent one time cell, shown separately for each rat. The correlation is significantly above zero for rats 2, 3, and 4. (**c**) Estimated trial variability $\sigma_t$ as a function of the peak firing time $M$. The line shows the best-fitting regression (see text for details).

The online version of this article includes the following figure supplement(s) for figure 4:

**Figure supplement 1.** Additional example time cell fits.

(*Figure 2c*); every other positive real number is also possible. We considered two alternative forms of distributions for the locations of time field centers across the population ('Exponential Compression Model' and 'Weibull Compression Model', *Table 1*).

The Main Model provided the best quantitative description of the data (see *Table 1*). Accordingly, the Results section focuses on the parameters from the Main Model with the results of alternative models noted where appropriate. Note that while the results are presented sequentially from the trial level to the population level, the parameters of each model were estimated using data from all levels of description simultaneously. Although the Bayesian analysis returns posterior distributions for all of the parameters it estimates, to test hypotheses about relationships between parameters we simply considered the mean of the posterior distributions as the model's estimate and performed inferential statistics on those means.

## The representation of time varies from trial to trial and the variance increases with the delay

Example fits of individual cells across trials for the Main Model can be found in *Figure 4a* (see also *Figure 4—figure supplement 1*). It is apparent that firing patterns change from trial to trial. The model captures trial variability by providing trial-level estimates of $\mu_i$ on each trial $i$ and by quantifying the across-trial variability for each cell *via* $\sigma_t$.

There are two novel findings reported in this section. First, the model was able to capture the significant variability in time field locations across trials. Model parameters were correlated with the timing of events at the beginning of the delay for each trial. This suggests that at least some part of the measured across-trial variability resulted from the variability in the behaviors at the beginning of the trial. Second, we found that the across-trial variability changed systematically as a function of the corresponding time field peak. This suggests that compression observed in previous studies of time cells is probably at least partially attributable to a trial averaging artifact. As we will see below, there is also robust evidence for compression that is not attributable to a trial averaging effect.

### Variability in the timing of events early in the trial accounts for some neural trial variability

The Main Model revealed a high degree of variability in the estimated time field location from trial to trial. *Figure 4a* plots representative single units with trial-by-trial model fits for cells that fired at different times during the delay interval. One factor that could account for the observed trial variability is rats' behaviors near to the initiation of the delay. After studying the stimuli for 3 s, animals were allowed to enter the treadmill, reach the end of the treadmill and start running on the treadmill. For each trial, the start of the delay was defined as the time when the animals first broke the laser beam pointing at the middle of the treadmill. However, it is possible that time cell sequence could be initiated by other events. Therefore the time gap between each event, which varies from trial to trial, could impact the initiation of the time cells sequences and result in an imperfect match to the clock time we measured. For some time cells, we found the time difference between the rat entering the treadmill and breaking the laser beam was positively correlated with the location of time field peak from trial to trial. The Pearson correlation score for each time cell is plotted in *Figure 4b*. The effect was particularly significant for rats 2, 3, and 4. We performed Bayesian one sample t-test using JASP (*JASP Team, 2022*) and found the mean correlation scores were significantly higher than 0 ($BF_{10} > 100$ for all three rats). Note we did not observe this effect for all animals as the same Bayesian one-sample t-test came out mildly in favor of the null hypothesis ($BF_{10} = .64$) for rat 1. We conclude that estimates of time field location across trials are a meaningful estimate of variability across trials and at least some of the variability can be accounted by the behavioral variability at the beginning of each trial. Because not all animals showed correlations with other events, and because the flag associated with breaking the laser beam is free of measurement error, we continue to measure time relative to the event of the animal breaking the laser beam on the treadmill.

Additionally, the effect of the initial event variability, quantified in the form of Pearson correlation value, decreases for time cells with centers later in the delay for rats 2, 3, and 4 (*Figure 4b*). To quantify this observation, we divided time cells into "early" *vs* "late" groups based on whether the average time field peak $M$ was less than or greater than 2 s. The correlation was significantly higher for the early group as assessed by a Bayesian two-sample t-test over the null hypothesis that correlation in early group ≤ late group ($BF_{10} > 10$ for rat 2 and 4; $BF_{10} = 3$ for rat 3).

### Estimated across-trial variability changes with delay

The Main Model separately estimates the within-trial variability—$\sigma_w$, the width of the time field within trial—and across-trial variability, $\sigma_t$. Discussion of within-trial variability is postponed to the next subsection. For each time cell $\sigma_t$ quantifies how much the time field shifts across trials. Although the logarithmic compression hypothesis makes quantitative prediction about time fields within a trial (*Equation 3*; *Equation 4*), it is agnostic about the question of how time is represented *across* trials. Nonetheless, it is important to quantify across-trial variability in the attempt to isolate the effect of within-trial variability on trial-averaged time fields. The model identified substantial across-trial variability. The median value of $\sigma_t$ was 0.69 s. Forty-five out of 131 time cells showed an across-trial

standard deviation of more than 1 s. The model's estimate of across-trial variability was larger than its estimate of the width of the time field for a substantial majority of time cells (93/131, $\chi^2(1) = 22.3$, $p < .001$).

### Across-trial variability changes over the delay interval

There was also a systematic change in across-trial variability as a function of the peak location (*Figure 4c*). There are a number of uninteresting factors that could affect this quantitative relationship. For instance, for a time cell with peak near the edge of the delay interval, it would be more difficult to estimate a large value of $\sigma_t$ because the distribution would be cut off by the edge of the delay. With this caveat in mind, we applied regression analysis of trial variability $\sigma_t$ as a function of the time field peak $M$, including an intercept, linear, and quadratic terms. All three coefficients were significant (intercept, mean ± SE: $-0.33 \pm 0.06$, $p < .001$; slope: $0.84 \pm 0.05$, $p < .001$; quadratic term: $-0.11 \pm 0.01$, $p < .001$). We also applied Bayesian linear regression and found that the model with both linear and quadratic term was the best model ($BF_M > 100$ over the null, the linear term only, and the quadratic term models). We conclude from this that variability in the location of the time field across trials varies systematically with the average location $M$.

Because variability in the location of a time field across trials is substantial and varies systematically with time within the delay, quantitative estimates of time field width as a function of time field center should control for this trial variability. Fortunately, the hierarchical Bayesian analysis provides separate estimates of within- and between-trial variability. We present results for time field width isolated from trial averaging effects in the next subsection.

## Hippocampal time cells form a logarithmically compressed timeline of the past

After accounting for across-trial variability, we are now able to study the form of temporal compression of the time code within a trial. If populations of time cells represent a compressed timeline, we should observe fewer units with larger values of $M$ and the time field width $\sigma_w$ should increase with $M$. This would mean that the temporal resolution carried by the population is reduced as the event fades into the past. If the population is compressed logarithmically, this makes two quantitative predictions. First, time field width $\sigma_w$ should increase linearly with the delay $M$ (*Equation 3*). Second, time field peaks $M$ should be distributed according to a power-law with an exponent of 1 (*Equation 4*).

### Time field width increases linearly with the delay

*Figure 5b* plots within-trial time field width $\sigma_w$ as a function of the peak location of the time field $M$ for each time cell. Examination of this plot suggests that time field width increases with delay, even after taking across-trial variability into consideration. To quantify this relationship, we fit a regression analysis of time field width $\sigma_w$ as a function of time field peaks $M$ time field width including constant, linear, and quadratic terms. Only the the linear term reached statistical significance ($0.14 \pm 0.05$, $p = 0.01$, $R^2_{adj} = 0.24$). The intercept ($0.11 \pm 0.06$, $p < .07$) and quadratic ($-0.007 \pm 0.008$, $p > 0.4$) terms were small and neither reached traditional levels of significance. A Bayesian linear regression analysis showed the best model to be one with only the linear term ($BF_M = 9.756$ over null, quadratic term only or quadratic plus linear terms models). The reliable linear term indicates that time field width increases with the peak time even after controlling for trial-averaging effects.

Logarithmic compression predicts a precisely linear relationship between within-trial time field width $\sigma_w$ and peak time $M$. While there was no evidence for a quadratic term, the intercept did approach significance. There is no question, however, that the curve cannot literally go to the point (0,0). For time cells to peak at $M = 0$ it would require that hippocampal time cells respond instantaneously, which is not physically possible. Moreover, it would be impossible to estimate a $\sigma_w$ approaching zero—it makes no sense to estimate a standard deviation without more than one spike. As it is, even if the intercept had reached significance at the upper limit of the confidence interval we observed (on the order of a few hundred ms), we would still conclude that the intercept is small in magnitude relative to the range of delay times exhibited by time cells in this experiment.

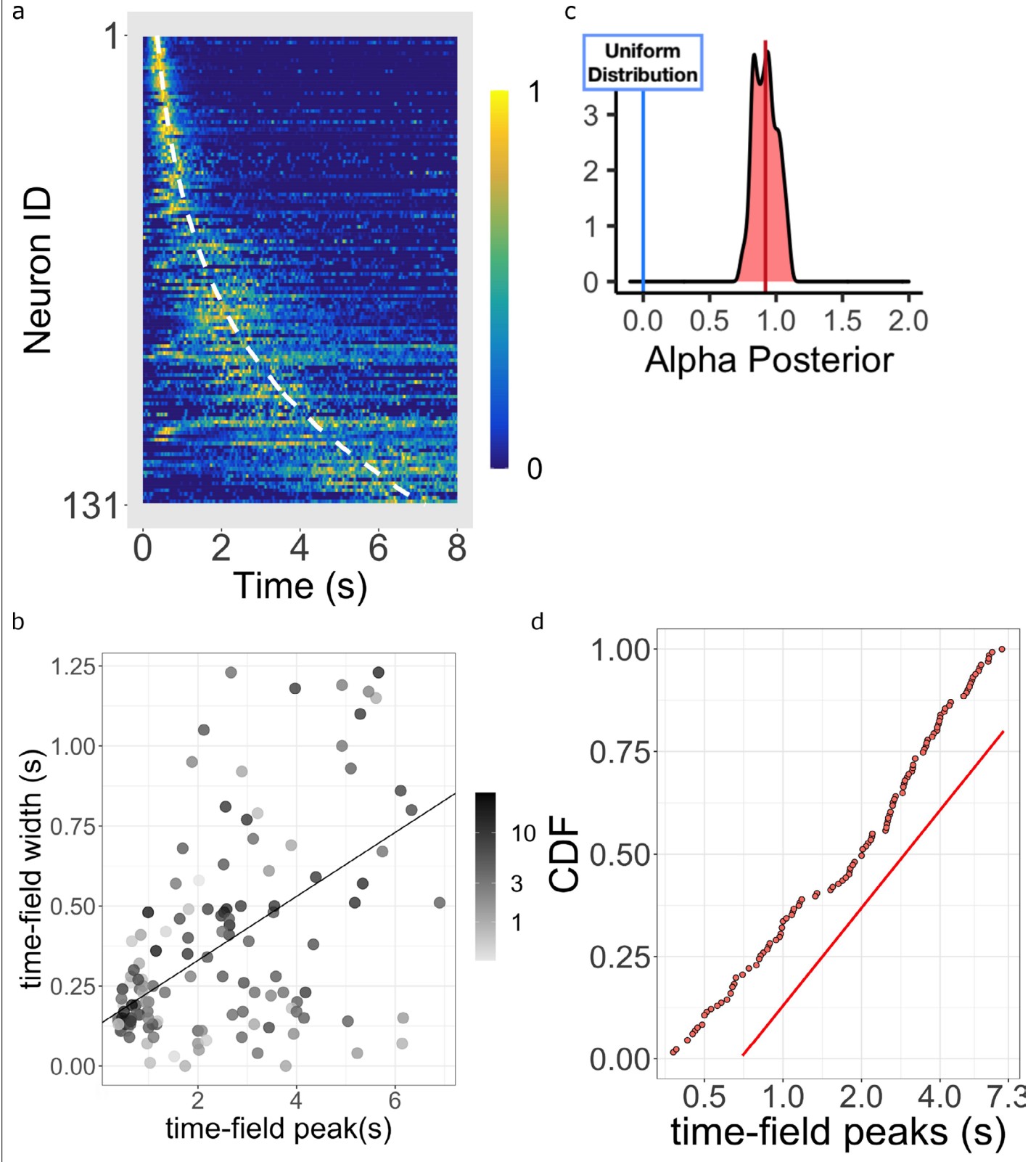

**Figure 5.** Log-compressed timeline. (**a**) Normalized firing rate of time cells sorted by peak time. Yellow indicates high activity and blue indicates low activity. The dashed white line shows the peak times that would be expected if time cells uniformly sampled $\log$ time. Equivalently, the dashed white line shows the peak times that would result if the probability of observing a peak time $M$ went down like $M^{-1}$ corresponding to $\alpha = 1.0$. (**b**) The width of the within-trial time field as a function of the peak firing time. The line shows the corresponding linear fit (see text for details). (**c**) The posterior

*Figure 5 continued on next page*

*Figure 5 continued*

distribution of the power-law exponent parameter $\alpha$. The blue line on the left marks zero, corresponding to a uniform distribution. The red line marks the mean (.93) of the posterior. (**d**) Cumulative distribution function (CDF) of time field peaks $M$ plotted on a $\log$ scale. The CDF function forms a straight line only when $\alpha = 1$, corresponding to logarithmic compression of the time axis. The straight red line has the same slope as the best fitting linear regression function to the CDF and is included to facilitate visual inspection of the cumulative function.

The online version of this article includes the following figure supplement(s) for figure 5:

**Figure supplement 1.** Additional analysis comparing the maximum likelihood fits and the hierarchical Bayesian fits.

## Time cell peaks are distributed according to log compression

*Figure 5a* shows the firing rate, averaged over trials, for each of the time cells sorted by their estimated peak time $M$. Consistent with many prior studies of time cells (e.g. *Mau et al., 2018*; *Cruzado et al., 2020*), the central ridge appears curved. The curvature is a sign of a non-uniform distribution of time field peaks $M$; if $M$ was uniformly distributed across time, one would expect to see a straight line when the data is plotted in this way. As observed in prior studies, the shape of the curvature appears consistent with logarithmic compression. The dashed white line on *Figure 5a* shows the curve that would be expected if the time fields were distributed uniformly over $\log$ time. The hierarchical Bayesian model enables us to quantify this relationship more precisely.

The Main Model estimates the distribution of time field peaks across the population as $M \sim M^{-\alpha}$ (see *Equation 10* and methods for more details). When $\alpha = 0$, the peak of time cells should be distributed evenly throughout the delay, showing no temporal compression. For $\alpha > 0$ the representation is compressed. Crucially, $\alpha = 1.0$ indicates that the compression is exactly logarithmic (*Equation 4*).

*Figure 5c* plots the posterior distribution of $\alpha$. This distribution is centered around 1 (mean = 0.92 with 95% credible interval [0.73, 1.11]). Furthermore, about 54% of the posterior density is between the value of 0.9 and 1.1. This posterior conclusively argues against uniformly distributed time cells as it excludes zero by approximately five standard deviations. It also argues strongly against other 'simple' values for $\alpha$ that one might have considered reasonable a priori, including 1/2, 3/2, and 2. To informally assess how well the distribution of $M$ approaches $\alpha = 1$, *Figure 5d* plots the cumulative distribution function (CDF) of time cell peaks $M$ on a log axis. If $\alpha$ were 1.0, the CDF would appear as a straight line. Conversely, $\alpha > 1$ would result in a sub-linear CDF curve and $\alpha < 1$ would result in a super-linear CDF curve. It should be noted that the estimates of $M$ for different time cells are not independent of one another in the hierarchical Bayesian analysis. However, one also finds similar results from traditional estimate of time fields where each cell's firing field is estimated independently (*Figure 5—figure supplement 1d*), at least considering fields that peak after a few hundred ms.

## Alternative models support the conclusion of a logarithmic timeline

The conclusion that time cell sequences are logarithmically-compressed (*Equation 3*; *Equation 4*) was buttressed by additional analyses testing variants of the Main Model. Similar conclusions about the representation of time are reached using all of the alternative models considered (*Table 1*).

The Main Model assumes that the distribution of time field peaks follows a power law. Although the posterior of the exponent $\alpha$ was tightly clustered around 1, it is possible that the distribution actually follows some entirely different equation. Although it is impossible to test every possible equation, we chose two well-known alternative skewed distributions, exponential and Weibull. These distributions provided much worse quantitative fits to the data than did the Main Model (*Table 1*). Nonetheless, one can still assess whether within-trial time field width $\sigma_w$ increases linearly with peak time $M$ (*Equation 3*). A linear increase was obseerved for both the Exponential Compression Model and the Weibull Compression Model (both $BF_M > 100$).

Another alternative model, the Log-normal Time Field Model, assumes that time field shape is log-normal instead of Gaussian at the trial level. This model, which produces skewed time fields, described the data a bit worse than the Main Model (*Table 1*). Nonetheless, time field width increased linearly with peak time in the Log-normal Time Field Model (*Equation 3*) and $\alpha$ was close to 1.0. A linear regression of width of time fields onto the peak time found the linear term reached statistical significance ( $.08 \pm .024$, $p < 0.001$, $R^2_{adj} = .085$). The same is true with Bayesian regression analysis: $BF_M = 58.54$ over null model. In addition, the posterior of the power-law scaling parameter $\alpha$ for the

Log-normal Time Field Model was centered around 1 (mean = .96 with 95% credible interval [0.77, 1.15]).

The Main Model allowed the time field center to vary across trials. The Alternative Trial Vary Model kept time field centers fixed across trials but allowed the width of time fields to vary across trials. Although the Alternative Trial Vary Model described the data much less well than the Main Model (*Table 1*), the parameters from this model were still consistent with the hypothesis of logarithmic compression of time fields. Time field width increased linearly with time field center ($BF_M > 100$) and the posterior of $\alpha$ was centered around 1 (mean = 1.01 with 95% credible interval [0.83, 1.15]).

The foregoing analyses demonstrate that the key predictions of logarithmic compression do not depend critically on any of the assumptions of the Main Model. *Figure 6a–b* provides an additional way to informally assess how well the logarithmic hypothesis describes the firing of hippocampal time cells independent of any assumptions of the hierarchical Bayesian method. In *Figure 6b*, the firing of each time cell is shown as a function of log time. If the time code for hippocampal time cells was logarithmically compressed, the peak would appear here as a line, indicating that the centers are uniformly distributed as a function of log time.

## Time cells early and late in the delay show similar compression

One possible concern is that hippocampal time cells are responding to some variable that changes over the delay. While it is in principle difficult to assess this question as we cannot measure every possible variable that could be of interest to the animal, we can rule out confounds with observable behavior. In this experiment, animals were not perfectly still as they ran on the treadmill over the 8 s delay. However, variability in position, running speed, and angular acceleration was much larger early in the delay as the animal moved towards the edge of the treadmill. Variance in these variables decreased dramatically after about 2 s (see Supplementary section S1). To assess the concern that the apparent logarithmic compression over the 8 s delay was an artifact of behavioral changes during that interval, we sorted time cells into two distinct groups with peak times before and after 2 s. Fortuitously, these groups included about the same number of time cells; the early group included 63 time cells and the late group 68. Both groups showed evidence of temporal compression when considered separately.

If the early and late time cells were two distinct groups, it is possible that this could give rise to an artifactual increase in time field width with delay. For instance, suppose that the width of time fields is constant within each group but larger for time cells in the later group. However, contradicting this artifactual account, time field width increased with peak time for both early and late time cells. Bayesian linear regression analysis showed evidence in favor of a linear model over the null model for both the early group ($BF_{10} > 100$) and the late group ($BF_{10} = 2.5$).

Similarly, perhaps the distribution of time field peaks observed in the Main Model is an artifact of averaging over early and late groups. Perhaps time field peaks are distributed uniformly within both the early and late groups. Additional analyses falsified this hypothesis. The distribution of time field peaks was non-uniform within each group and there was no evidence that the two groups were sampled from different distributions. A KS test against a uniform distribution sampled over the range of time field peaks in each group rejected the uniform hypothesis for both early, $D(63) = .307, p < .001$, and late group, $D(68) = .305, p < .001$. To assess whether the distributions of early and late time field peaks were different from one another, we normalized peak times within each group such that each peak time was between 0 and 1. Comparing the distributions of relative peak times between early and late groups, we could not rule out the hypothesis that they were sampled from the same distribution, $D(63, 68) = .090, p > .5$.

To summarize, despite dramatic differences in behavioral variability early and late in the delay, we found no evidence suggesting that early and late time cells had different properties from one another nor that either group was qualitatively different from the population as a whole. Coupled with dramatic changes in behavioral variability during the early and late periods of the delay, this argues against the hypothesis that the empirical results arguing for logarithmic compression are artifacts of behavioral variability during the delay.

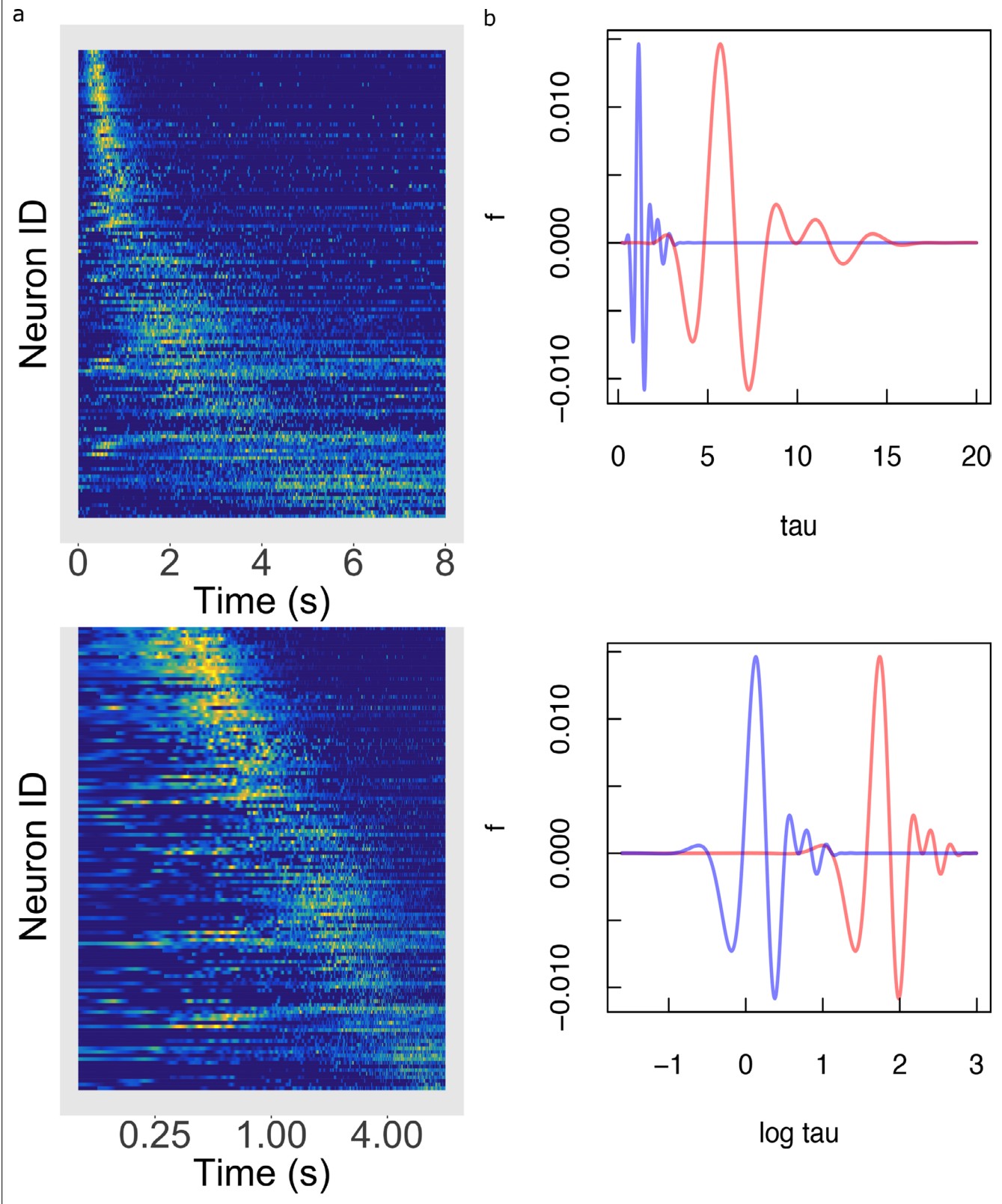

**Figure 6.** Temporal rescaling translates a logarithmically compressed time line. (**a**) Normalized firing rate of time cells sorted by peak time on a linear time scale (top) and on a logarithmic time scale (bottom). Yellow indicates high activity and blue indicates low activity. When plotted on a linear scale, the sequence of time cells appears curved and widens with the passage of time. In contrast, when plotted on a $\log t$ axis time cells appear to uniformly tile the delay. (**b**) Two functions of time related to one another by a constant scaling factor $f(t)$ and $f(at)$. (Top) The two functions are plotted as a

*Figure 6 continued on next page*

*Figure 6 continued*

function of linear time $t$. (Bottom) The two functions plotted as a function of $\log t$. Rescaling $t \rightarrow at$ simply results in a translation of $f$ along the $\log t$ axis.

## Discussion

The present study analyzed a group of hippocampal CA1 time cells with a hierarchical Bayesian model that describes the firing patterns simultaneously at the individual trial level, the individual cell level, and the population level. These analyses confirmed two quantitative predictions that follow from the hypothesis that hippocampal time cells evenly tile the $\log$ time axis (*Equation 1*). First, analysis at the individual cell level, with variability across trials removed, showed that within-trial time field width increased linearly with delay (*Equation 3*). Second, the distribution of time field peaks followed a power-law with exponent close to one (*Equation 4*). These findings were robust to many choices of the analysis methods. Taken together, these findings show strong evidence that time cells maintain a logarithmic timeline (*Equation 1*) within trial.

In addition to these findings at the cell and population levels, there was also substantial variability in time field location across trials. At least some of this variability was attributable to variability in the time of events at the beginning of each trial as the animal advanced onto the treadmill and began to run. However measurable behavior (position, velocity, acceleration) was relatively consistent after about two seconds into the trial. To address the concern that the temporal compression in hippocampal time cells is an artifact of behavior, we tested for temporal compression within the population of cells that fired earlier than 2 s and those that fired after 2 s. Both populations showed evidence for temporal compression—wider fields later in the delay and an overrepresentation of time points early in the delay—taken separately. This set of results suggests that the findings suggesting that time cells form a logarithmic timeline are not likely to be an artifact of variability in behavior that is correlated with time within the trial.

### Compressed representations of time and space in hippocampus, entorhinal cortex and other brain regions

Many other studies of time in the hippocampus and other brain regions show a broad spectra of timescales across neurons and evidence of compression at the population level. That is, many prior studies show a graded heterogeneity of time scales across units and time scales close to zero are overrepresented relative to time scales representing times further in the past. Representations of spatial location in the hippocampus and related regions could be logarithmically compressed as well. This suggests a common computational framework for representation of many kinds of continuous variables in the hippocampus and elsewhere.

#### Sequentially-activated time cells in many brain regions

A growing body of work suggests that the brain uses a graded heterogeneity of time scales to represent the past. However, it is not clear if Weber-Fechner compression is a general property. Many previous papers have shown compression of hippocampal time cells in rodent hippocampus consistent with log compression (e.g. *Kraus et al., 2013*; *Mau et al., 2018*; *Taxidis et al., 2020*). In addition, a qualitatively similar compression has been reported for sequentially activated cells in rodent striatum (*Mello et al., 2015*; *Akhlaghpour et al., 2016*; *Parker et al., 2022*), rodent MEC (*Kraus et al., 2015*), rodent mPFC (*Tiganj et al., 2017*), monkey caudate nucleus and DLPFC (*Jin et al., 2009*), monkey lPFC (*Tiganj et al., 2018*; *Cruzado et al., 2020*) and monkey hippocampus (*Cruzado et al., 2020*) during a broad variety of behavioral tasks.

#### Graded heterogeneity of decay and ramp rates in entorhinal cortex

In addition to sequentially-activated time cells, ramping and decaying activity in the entorhinal cortex (EC) can be used to decode time since a triggering event. Rather than firing sequentially with graded heterogeneity of peak times, cells in EC, which have been referred to as temporal context cells, appear to change their firing rate gradually following an exponential function as the triggering event recedes into the past. Recording from rodent LEC in a spatial exploration task, *Tsao et al., 2018* observed cells that gradually changed their firing rate with a wide range of time constants spanning

up to several minutes. Graded heterogeneity in time constants of decay (or ramp) across cells allows for the decoding of time since an event over a wide range of time scales. *Bright et al., 2020* observed similar results in monkey EC during a free viewing task with cells triggered by onset of an image and decaying with time constants ranging from a few hundred ms up to several s.

Entorhinal neurons that are not typically understood to be coding for time also show a graded heterogeneity of time constants. *Dannenberg et al., 2019* recorded from speed cells in rodent MEC. During exploration speed changes continuously; that study asked over what time scale filtering the speed signal provided the best account of each speed cell's time course of firing. Rather than finding a single value, different speed cells appeared to filter the speed signal with a wide range of time constants. The distribution over cells overrepresented times near zero and showed a long tail, suggesting a compression of the time axis. Although there are many possible biological mechanisms for graded heterogeneity of time constants, slowly-inactivating currents in EC observed in slice experiments (*Egorov et al., 2002*) are an attractive candidate for the graded heterogeneity of time constants in EC (*Tiganj et al., 2015*; *Liu et al., 2019*).

## Graded heterogeneity of time constants in cerebellum and other cortical regions

Work in the cerebellum and cortex outside of the MTL also shows graded heterogeneity of time constants. For instance, graded temporal heterogeneity in the cerebellum has been argued to result in temporal basis functions to represent what happened when in the past (*Barron et al., 2020*; *Kennedy et al., 2014*; *Raymond and Medina, 2018*; *Cavanagh et al., 2020*; *Guo et al., 2021*). Merchant and colleagues recorded neurons in the medial premotor cortex during rhythmic tapping behaviors with a variety of speeds (e.g., *Merchant and Averbeck, 2017*). They observed that some neurons are sensitive to the period of the interval. They observed a graded heterogeneity of preferred intervals across neurons. These same neurons showed gradual ramps or decays that extended several seconds. In addition, a wide variety of studies have shown that the firing rate of neurons regions within the monkey PFC carry temporal information during working memory tasks (e.g. *Machens et al., 2010*). For instance *Cueva et al., 2020* measured the dimensionality of the neural space as time elapsed. The dimensionality increased gradually, consistent with graded heterogeneity in time constants. Morever, the rate at which the dimensionality increased slowed with the passage of time, consistent with temporal compression reported for hippocampal place cells.

Work from the cerebellum shows evidence for a broad distribution of time scales and at least some evidence that the time constants are distributed along a logarithmic scale. *Guo et al., 2021* reported that unipolar brush cells in the cerebellum form a logarithmically-compressed temporal basis set in a slice preparation. They were able to attribute slow time scales in the slice to heterogeneity in metabotropic glutamate receptors.

It is theoretically important to systematically evaluate whether the distribution of time constants across regions and tasks shows the same quantitative distributions. This would require careful experimentation. Because the experimental challenges of estimating time are greatest for times near zero, it may be preferable to conduct this experiment over time scales greater than a few seconds. It has been argued that hippocampal time cells at least fire sequentially over several minutes (*Shikano et al., 2021*; *Liu et al., 2022*).

## Compressed spatial representations in hippocampus and EC

There is evidence suggesting that place fields cluster relative to landmarks in the environment, such as the beginning of a path (*Bjerknes et al., 2018*; *Sheehan et al., 2021*) and edges of an open-field environment (*Harland et al., 2021*), a result analogous to the compression of time observed here. In order to make a clear comparison between temporal representations and spatial representations, it is essential to establish the 'zero' of the spatial coordinate system. That is, populations of time cells appear to code for the time between a particular event and the present. However, when an animal is in any specific location in a spatial environment, many different landmarks are at different distances and headings. In order to directly compare time to space, it is essential to establish experimental control over the place code by moving a landmark in the environment and observing that the place cells move with the landmark. *Sheehan et al., 2021*, following a method introduced by Gothard and colleagues (*Gothard et al., 1996*; *Gothard et al.,*

*2001*), systematically manipulated the location of a reward box along a linear track. As observed by *Gothard et al., 1996*; *Gothard et al., 2001*, many hippocampal place fields moved with the box, coding for distance traveled since leaving the box. Consistent with the results from time cell experiments, place fields tiled the distance around the box smoothly. Moreover, place fields were more numerous and more narrow closer to the landmark controlling their firing than further away from the landmark (*Sheehan et al., 2021*). It is not currently known whether the form of the compression of the place code is also logarithmic, but the results are at least qualitatively similar to those observed for time cells (see also *Bjerknes et al., 2018*).

It has long been appreciated that the spatial frequency of grid cells falls into discrete groups that are anatomically organized (*Stensola et al., 2012*). Interestingly, the relative ratio between the spatial frequency of adjacent clusters of grid cells is constant, as predicted by a normative account of spatial coding (*Wei et al., 2015*). Logarithmic compression also implies that the ratio between adjacent time scales are constant (i.e. *Equation 2* implies that $\frac{t_{n+1}}{t_n} = b$). That is, both spatial frequency of grid cells and temporal receptive fields for time cells evenly tile a logarithmic axis (equivalently, form a geometric series). The major differences are the parameter that controls the ratio between adjacent neurons and the number of cells that share each value. Whereas time cells appear to smoothly cover the $\log$ time axis, many grid cells cluster at a few evenly-spaced locations on the $\log$ spatial scale axis. Whereas the ratio between adjacent time scales is close to one, the ratio of the spatial frequency of one cluster of grid cells to the next is much larger. It is an interesting theoretical problem to understand why the brain has chosen distinct but closely related coding schemes for time (and perhaps also space) in the hippocampus and spatial frequency in the grid cell system. Perhaps a solution to this problem would also make sense of the firing of grid cells from MEC when recorded in a time cell experiment (*Kraus et al., 2015*) and the observation of time-like sequences in MEC during pauses in spatial exploration (*Heys and Dombeck, 2018*).

## Implications of logarithmically compressed time for computational cognitive neuroscience

Knowing the mathematical form of the timeline represented by hippocampal time cells allows us to infer many properties of the population. *Figure 6* illustrates a property called 'scale-covariance' that is a consequence of a logarithmically-compressed timeline. The top panel of *Figure 6b* shows a pair of functions of time that are rescaled. To get a physical intuition to time rescaling, imagine that you repeat a behavorial experiment on an animal, but in the repetition of the experiment every time interval is changed by the same factor. We would say that the second version of the experiment is a rescaled version of the first. When plotted on a log axis in the bottom panel of *Figure 6b* the two curves have identical shape, but are shifted relative to one another. More formally, we can say that time rescaling a function results in translation on a logarithmic scale. This is true, for all possible functions and all possible scaling factors, for the simple reason that $\log(at) = \log a + \log t$. Rescaling time—taking $t$ to $at$—translates the neural representation, shifting time cell indexes by a factor proportional to $\log a$. This property of a logarithmically-compressed timeline is referred to as scale-covariance. It can be shown that only logarithmic functions give rise to this form of scale-covariance.

The scale-covariance of the neural representation should not be confused with scale-*invariance*. A scale-invariant system is completely unaffected by temporal rescaling. As we will see below, it is straightforward to build scale-*invariant* computational models from scale-*covariant* neural representations. There is, however, also evidence for scale-invariant neural representations of time. For instance, *Mello et al., 2015* observed cells in the striatum that fire sequentially in a fixed interval operant conditioning task. In the fixed interval procedure, the first response after a particular interval has past is reinforced. Animals learn to respond at appropriate times. *Mello et al., 2015* observed that when the fixed interval changed, the sequences were stretched in time. If we examined the population at the moment just before the interval ends, we would find the same pattern of firing across neurons regardless of the duration of the interval. That is, the pattern of firing across the population is invariant to rescaling time in the experiment. Neural responses that respond to serial or ordinal position even as the timing of events is rescaled are also scale-invariant (e.g., *Crowe et al., 2014*; *Machens et al., 2010*).

## Logarithmic timelines and scale-invariant perception

One can compute scale-invariant quantities from a scale-covariant neural representation. For instance, consider *Figure 6b*. When plotted as a function of log time the rescaled functions have the same standard deviation. The standard deviation computed from the scale-covariant representation is scale-invariant. This property of a logarithmically-compressed timeline is closely related to the 'scalar timing' observed behaviorally (*Gibbon, 1977*; *Rakitin et al., 1998*; *Lejeune and Wearden, 2006*) and neurally (*Crowe et al., 2014*).

When integrated into deep neural networks, scale-covariance can be exploited to build networks for perception that naturally generalize to rescaled signals. *Jacques et al., 2022* trained a deep network with logarithmically-compressed time cells at each layer to learn to classify auditory patterns (*Jacques et al., 2021*). After being trained to classify digits spoken at a normal rate, the network built from logarithmically-compressed time cells recognizes digits spoken much more slowly or quickly than the training examples without retraining. Notably, it has been argued that time constants in the early auditory system have a logarithmic distribution (*Rahman et al., 2020*). *Jansson and Lindeberg, 2022* used a similar idea, building on longstanding theoretical work in the context of computer vision (*Lindeberg and Fagerström, 1996*; *Lindeberg, 2016*), to build networks for visual pattern classification that generalize to images that are larger or smaller than the training set. It has long been known the retinotopic coordinates in the early visual system compress distance from the fovea logarithmically (*Van Essen et al., 1984*).

The ubiquity of scale-covariant representations in the brain suggests that this choice for distributing receptors is a consequence of some general computational principle. Viewed from the perspective of efficient coding (*Brunel and Nadal, 1998*; *Wei and Stocker, 2015*; *Wei and Stocker, 2016*), logarithmic scales are a natural consequence of the expectation of power law statistics. If the probability of observing a stimulus of value $x$ goes down like $x^{-1}$, then a neural code in which receptive fields tile $\log x$ results in each unit firing equally often (*Wei, 2012*). Indeed, researchers have noted that many natural signals have long-range correlations (e.g. *Anderson and Schooler, 1991*). However arguments about the metabolic or coding efficiency of a population of neurons need not lead to the conclusion that logarithmic scales are a consequence of a prior belief in power law statistics. For instance, in the presence of white noise, a logarithmic temporal code equalizes the mutual information between adjacent receptors (*Shankar and Howard, 2013*; *Hsu and Marzen, 2020* see also *Howard and Shankar, 2018*).

## Logarithmic timeline and temporal context in retrieval of episodic memory

In episodic memory, scale-covariance can make sense of a number of an otherwise puzzling finding about epsodic memory in humans. Human episodic memory—the vivid recollection of a specific event from one's life—has long been thought to depend on recovery of a prior state of a gradually changing state of spatiotemporal context (*Tulving and Madigan, 1970*). The retrieved context hypothesis has led to a number of detailed computational models of human behavior (e.g. *Sederberg et al., 2008*). These computational models explain the contiguity effect in episodic memory—the finding that memory for a specific event spontaneously brings to mind events experienced at nearby points in space and time (*Healey and Kahana, 2014*; *Kahana, 2012*)—as a consequence of successful retrieval of a prior state of spatiotemporal context.

Remarkably, the contiguity effect in episodic memory is observed across a very wide range of time scales. For instance, in the free recall task recall of a word from a sequentially-presented list tends to be followed by recall of a word from a nearby position within the list (*Kahana, 1996*). In a typical free recall task in which the words are presented one per second, a robust contiguity effect is observed (*Kahana, 2008*). However, if one inserts a distractor-filled delay of sixteen seconds between each word, presumably clearing the contents of short-term memory between each list item, a contiguity effect is still observed (*Howard and Kahana, 1999*). If, at the end of an experimental session lasting tens of minutes, the experimenter asks the participant to recall all the words from all the lists, there is a contiguity effect *across lists*, despite the fact that the words from different lists are separated by hundreds of seconds (*Unsworth, 2008*; *Howard et al., 2008*). At even longer scales, the contiguity effect can be observed when stimuli are presented separated by an hour (*Cortis Mack et al., 2017*) or when participants recall news events extended over weeks and months (*Uitvlugt and Healey, 2019*).

The persistence of the contiguity effect in episodic memory over changes in the time scale of the experiment are a serious challenge for attractor models for retrieval of spatiotemporal context. Attractor dynamics of such a network would have to be invariant to rescalings of time. Time cells and place cells are an appealing candidate for a neural implementation of a representation of spatiotemporal context (*Howard et al., 2015*; *Eichenbaum, 2017*). Logarithmic compression of time—and the scale-covariance that follows from this property—offers a solution to this theoretical problem. Rescaling of time translates the covariance matrix between time cells (up to a constant factor) along the log time axis. If the covariance matrix can be shaped to form a line attractor (*Spalla et al., 2021*; *Zhong et al., 2020*; *Chaudhuri et al., 2019*) this would enable retrieval of logarithmically-compressed states of spatiotemporal context.

### What is the upper limit of the logarithmic scale for time?

Although time cell sequences may continue much longer than the eight second delay used in this experiments (*Shikano et al., 2021*; *Liu et al., 2022*), it is difficult to imagine that time cell sequences persist for hours and days. However, it is possible that the temporal code is logarithmically compressed, perhaps by means of distinct neural mechanisms, over time scales much longer than minutes. For instance, *Nielson et al., 2015* studied the pattern similarity across voxels in the hippocampus evoked by different real world memories. Pairs of memories were separated by objective distances in time and space. *Nielson et al., 2015* found that pattern similarity fell off as a function of log time extending up to weeks. Behaviorally, *Arzy et al., 2009* asked human participants to place real world autobiographical events on a timeline centered at the present. They found the accuracy for the time of past events decreased linearly with log-scaled time, suggesting a logarithmically-compressed temporal representation over time scales up to decades (see also *Peer et al., 2015*). Representational drift—slow changes in the pattern of active neurons when a particular stimulus or experimental setting is repeated—is a possible candidate to play the role of spatiotemporal context over autobiographical time scales (*Ziv et al., 2013*; *Rubin et al., 2015*; *Cai et al., 2016*; *Mau et al., 2018*; *Mau et al., 2020*).

## Materials and methods
### Experiment and electrophysiology
#### Subjects

Subjects were four male Long-Evans rats (Charles River) weighing between 350 and 450 g and between the ages of 6 months to 1.5 years for the duration of the experiment. Animals were single housed and maintained on a 12 hr light-dark cycle (lights on at 8:00 AM) for the duration of the experiment. All behavioral training and experimentation were performed during the light phase. Animals were given ad-libitum water and maintained at a minimum of 85% of their ad libitum feeding body weight during all behavioral training and testing. All procedures were conducted in accordance with the requirements set by the National Institutes of Health, and were approved by the Boston University Institutional Animal Care and Use Committee (BU IACUC).

#### Behavioral apparatus

The behavioral apparatus consisted of a custom-built 355 cm long by 7.5 cm wide circular track with an integrated treadmill using commercially available parts (Columbus Instruments). The treadmill had walls funneling into a small exit to ensure the animals' head position was fixed for the duration of the treadmill run. At the end of the treadmill existed a 20 x 20 cm square platform onto which the test objects were placed. The track was elevated 95 cm above the floor, positioned at least 20 cm away from all walls but in a position in which various distal visual cues were available to the rat. The maze contained two automatic doors, one at the front of the treadmill and one on the return arm that was controlled by infrared beam breaks (Arduino microcontroller, hardware from Adafruit industries).

#### Training procedure

Rats were trained in a similar manner to previous experiments (*Robinson et al., 2017*). Briefly, rats were initially trained to run in one direction around the maze for froot-loop reward. Once rats reliably looped, they were shaped to retrieve the reward hidden in a flower pot each lap (roughly 4 days). Then, rats were trained to sample one of two test objects before running through the treadmill

platform and then choosing the matching one of two terra cotta pots discriminable by scent and digging media (*Keene et al., 2016*). The terra-cotta pots were always placed side-by-side on the platform in a pseudorandomized position. A choice was determined to be made once the animal disturbed the surface of the media at which point the opposite pot was immediately removed at this and all subsequent stages of behavior. One crushed froot loop was added to each pot at the beginning of each day, and every 7–9 trials neither pot contained a food reward, and the reward was given after 1 second of digging to prevent the possibility of reward scent guiding behavior. This shaping was performed by initially running each object-pot pair in large blocks of trials (>10 consecutive trials) and progressively weaning down to random presentation over the course of 2 weeks. Once animals were performing at >80% discrimination, two new objects were added one at a time to the study set using the same progressively shrinking block design. Once animals were able to discriminate between the 4

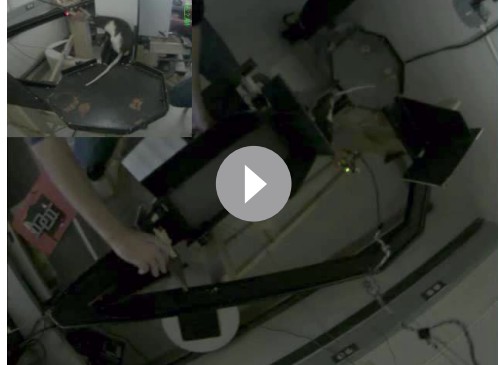

**Video 1.** Video of example trials Three consecutive trials from example session (Rat 1). An Overhead camera tracked the position of the rat on the maze and on the treadmill, whereas a second camera (inset) positioned at the reward objects allowed us to precisely score when the rat made a choice. Data were acquired at 30 frames per second at 640x480 pixel resolution (1/2 cm per pixel for overhead camera).

https://elifesciences.org/articles/75353/figures#video1

objects to retrieve reward from the correct terra-cotta pot of the two with >90% accuracy, a running delay was imposed. Initially, animals were trained to wait for a fraction of a second while the treadmill was nudged forward between the study and test phases. This treadmill delay was progressively increased until performance fell to between 70 and 80%, typically 8 seconds. Typically, this training schedule took roughly 2 months to train animals on the first pair of study objects, and then another 1.5 months for rats to perform with 4 study objects and with a sufficiently long treadmill delay.

## Surgery

Anesthesia was induced via inhalation of 5% isoflurane (Webster Veterinary Supply) in oxygen, and then a stable plane was maintained at 1.5–3% throughout the entire surgery. Immediately following the induction, animals were injected with the analgesic Buprenex (Buprenorphine hydrochloride, 0.03 mg/kg i.m.; Reckitt Benckiser Healthcare), and the antibiotic cefazolin 330 mg/ml i.m.; West-Ward Pharmaceuticals. The rat was then fixed to a stereotaxic frame (Kopf). Craniotomies then were made above the dorsal hippocampus (dHPC) (AP-4.1, ML 3.0 mm), and the rostral lateral entorhinal cortex (LEC) (AP-6.8, ML 4.5). Six to eight sterile stainless steel screws were then fixed to the skull, and the remaining skull surface was covered with Metabond (Parkell, Inc). A hyperdrive consisting of 24 independently moveable tetrodes was then lowered into the craniotomy, and fixed to the skull using dental cement. Two ground screws were inserted above the cerebellum, and soldered to the ground leads on the electrode interface board during surgery. Each tetrode was composed of four 12 µM RO 800 wires (Sandvik Kanthal HP Reid Precision Fine Tetrode Wire; Sandvik). Tetrodes were plated with non-cyanide gold solution (Neuralynx) via electrolysis using a Nano-Z (White Matter LLC) in order to reduce impedance to between 180 and 220 kOhms. At the conclusion of surgery tetrodes were lowered 1 mm into dorsal cortex. Animals were given Buprenex and Cefazolin twice a day as needed for up to three days post-surgery.

## Data acquisition

All electrophysiological recordings were performed using a 96 channel multichannel Acquisition Processor (MAP) recording system (Plexon). Each spike channel was amplified 1000 x, then between 3 and 10 x depending on the region. Spike channels were manually referenced to a local tetrode with no apparent unit firing, bandpass filtered between 200 and 10 kHz, and digitized at 40 kHz. LFP signals were uniformly referenced to ground, amplified 2000 x, and bandpass filtered between 0.5 Hz and 300 Hz. Tetrodes were moved a minimum of 60 uM after each recording session to prevent resampling

the same units. Tetrodes were lowered a minimum of 12 hr before all recording sessions to ensure tetrode stability. The animal's position and behavior were tracked via two LEDs fixed to his hyperdrive on one overhead camera, and on one camera positioned behind the two terra-cotta pots each recording at 30 fps and about 2 pixels per cm. Cineplex Studio (Plexon) was used for capturing behavioral data, and Cineplex Editor (Plexon) was used to manually enter event markers and to verify animal position tracking. For position data, the LED fixed to the rat's implant was automatically tracked online in Cineplex. Missing data epochs larger than 1 s were manually interpolated, and those less than a second were linearly interpolated. Behavioral events were manually scored using a two camera system to enable frame-by-frame determination of behaviors (*Video 1*). Crucially, we hand scored the moment the rat placed his first paw on the treadmill as well as the moment he touched the closed door at the front of the treadmill. The treadmill was initiated via IR beam break positioned at the midpoint of the treadmill and was not hand-scored. The delay ended automatically and was signaled by simultaneous termination of the treadmill belt and opening of the exit door.

## Spike sorting and data processing

Spikes were assigned to individual units by offline manual clustering of waveform characteristics (valley amplitude, peak-valley, energy, principal components, and waveform width at half max). Tetrode localization was performed using histology, and further guided by the LFP theta amplitude and phase, and sharp-wave ripple amplitude. Only well-isolated units (Isolation Rating > 40 L-Ratio <0.001) were kept for further analysis. Behavior was monitored from an overhead camera and a second camera with a close-up view of the test pots (see above). Door movements, and treadmill onset and offset were automatically strobed into the recording amplifier. Test object sampling events were hand-coded using CinePlex editor (Plexon) using video taken at 30 Hz from the camera trained on the reward objects.

## Data analysis

The goal of the analysis is to find the parameters that best describe the properties of time fields from trial to the population level. First, we identify time cells using the standard methods (*Tiganj et al., 2018*) and systematically eliminate time cells near the event boundary. Then we apply five hierarchical Bayesian models to this population (*Table 1*). The Main Model allows us to test the logarithmic compression hypothesis directly. Although the Main Model is presented first to aid readers' understanding of the fitting process, we also considered alternative ways to characterize the properties of time fields at *every level* of the hierarchy and tested those alternative models following the same model fitting procedures.

## Selecting time cells as input to hierarchical Bayesian models

As a first pass to identify time cells to be passed on to the hierarchical Bayesian models, we used established methods reported in other papers (*Tiganj et al., 2018*; *Cruzado et al., 2020*). Additionally, the models aim to characterize the population as a whole to estimate the temporal compression, we exclude a small portion of neurons if their estimated peak firing is too close to the event boundaries (*Clauset et al., 2009*). Finally, to ensure that the variability across trials could be reliably estimated, we required that units considered in the hierarchical Bayesian analysis fired at least one spike on at least 20 trials.

## Prior methods for estimating time cells

The particular method we applied to identify time cells (*Tiganj et al., 2018*; *Cruzado et al., 2020*) is conservative in that it uses multiple tests to ensure that a particular unit reliably codes for time across trials and is meaningfully related to time within the delay period. It does not control for variability across trials nor to does it try to assess the properties of the population.

In order to be classified as a time cell, a unit had to satisfy several criteria. First, to exclude interneurons, we required the firing rate over all delay intervals had to be less than 5 Hz. Second, to be classified as a time cell, units had to show a reliable time field. This was quantified by comparing two models to describe the firing rate as a function of time. One model included a Gaussian-shaped time field, with a mean and a standard deviation, and a constant term. The other model only estimated a constant firing rate. Best-fitting parameters for each model were identified for each unit and each

model using scipy's basin-hopping method scipy.optimize.basinhopping with truncated Newton algorithm (arguments stepsize = 1000, niter = 500, method = TNC, use_jac = T, t=1). In order to pass this criterion, the log-likelihood ratio for the time field model compared to the constant model had to exceed 5.66. The two models differ by three degrees of freedom; 5.66 is the threshold needed to exceed the log-likelihood ratio test with three degrees of freedom at the $p < .01$ level. Additionally, different criterion options were tested and we found six more units (average time field peak at 1.12 s) would be admitted if the criterion is relaxed to $p < .05$. On the other hand, six units (average time field peak at 1.54 s) would be eliminated if the criterion is tightened to $p < .005$. Since only a small fraction of the population (6/159) is impacted by the choice of criterion and the impact seemed to be unbiased (median time field peak of the population is 1.22 s), we proceeded with the criterion of $p < .05$ following the previous examples (*Tiganj et al., 2018*).

To ensure that the units classified as time cells were reliable across trials, we required that the time field model was statistically reliable for both even and odd trials considered separately. To ensure that the method identified time fields within the delay interval, rather than monotonically ramping or decaying firing (or a time field centered outside the interval), we additionally required that the mean of the Gaussian firing field was within the delay.

## Setting the bounds for temporal compression estimation

Because the hierarchical Bayesian models we considered estimate the compression at the population level as they estimate the time field peak location for each cell and trial simultaneously, it is important to exclude time cells near the event boundary. Specifically with the power-law distribution, which we used to quantify the temporal compression at the population level, there exist some standard procedures to exclude extreme values (*Clauset et al., 2009*). *Clauset et al., 2009* argued that the chosen boundary should make the probability density function from the bounded data as close to the best-fit power-law distribution as possible. Following their advice, we performed Kolmogorov-Smirnov statistic (KS test) to measure the probability that the bounded data, estimated time field peaks from the maximum likelihood methods in our case, was sampled from the best-fit power-law distribution and chose the boundary parameters that produced the highest p-value. For the lower bound parameter (*min* parameter in the hierarchical Bayesian model), we searched the range from 100 to 400 ms with a 50 ms grid. For the upper bound parameter(*max* parameter in the hierarchical Bayesian model), we searched in the range of 6400–8000 ms with a 400 ms grid (5% of the overall trial-length 8000 ms). For each of the lower bound and upper bound pairs, we took the estimated time field peaks that fell in the range and estimated the corresponding power-law distribution (*Equation 10*) with the Bayesian model fitting method. The posterior of the compression parameter $\alpha$ is generated through the rStan package (*Stan development team, 2021*) with three independent Markov-Chain-Monte-Carlo (MCMC) chains (4000 warm-up iterations and 1000 post warm-up samples in each of the MCMC chains). Then we applied the KS test that compares the data against the best-fitting power-law distribution composed with the corresponding $\alpha$. The cumulative density function (CDF) of the bounded power-law is defined as:

$$P(x) = \frac{x^{1-\alpha} - \min^{1-\alpha}}{\max^{1-\alpha} - \min^{1-\alpha}} \tag{9}$$

In our KS test results, we found the pair with the highest p-value is (350ms, 7200ms) with a p-value of .855. We excluded any cells outside of the range (14/159) and applied this range limit for the compression estimation in the hierarchical Bayesian models fitting below. Although we ended up with one particular boundary pair, the estimated $\alpha$ distribution does not change much between different pairs in our results: after excluding some very early choices (100-200 ms) for lower bound and very late choice (8000 ms), power law distribution provides a reasonable fit from the remaining pairs: $p > 0.5$ based on KS test. Moreover, the corresponding estimated $\alpha$ posteriors are between 0.5 and 1.5 and always include 1.

## Hierarchical Bayesian estimation of time fields across trials and neurons

We tested five hierarchical Bayesian models for the data set (see *Table 1*). Here, we first describe the Main Model. A schematic illustration of the Main Model can be found in *Figure 2*. Following the Main Model, we describe alternative models where we scrutinize the specific model assumptions by testing

alternative hypotheses, one level at a time to isolate the impact. All of the models follow the same fitting procedures.

Bayesian model fitting procedures search the space of possible values of parameters through iterations to improve the likelihood of predicting the observed data. However, rather than finding the *best* fitting value, the program returns a posterior distribution that include a range of likely values for a given parameter. For expository purposes we discuss the specific hypothesis at each level in sequence, but the program fits parameters from all levels simultaneously. Each model has parameters that affect the likelihood of the observed data, aggregated across cells and trials. One only estimates hypotheses from fits of all levels together. Therefore all the alternative models adopt the assumptions of the Main Model *except* the specific assumption they aim to test.

The posterior distributions of estimated parameters were generated through the rStan package (*Stan development team, 2021*) with 8 independent Markov-Chain-Monte-Carlo (MCMC) chains (4800 warm-up iterations and 200 post warm-up samples in each of the MCMC chains). Stan returns the multiple levels simultaneously. This procedure returns posteriors of the likelihood of the data as well as posteriors for the parameters of the model.

## Main Model

The Main Model assumes a Gaussian-shaped time field, with the peak location at $\mu_i$ and width of $\sigma_w$, for each time cell (*Equation 5*). The location of the time field is assumed to shift across trials according to a normal distribution that centered at $M$ with standard deviation of $\sigma_t$ (*Equation 7*). The time field peaks $M$ are distributed according to the power-law across the population (*Equation 10*). Because we have described the Main Model at the beginning of the Results section and there are no amendments to the description of model assumptions at the trial level and the population level, here we continue to describe the Main Model at the population level.

The Main Model assumes that the population of time cells is distributed according to power-law function: $y = x^{-\alpha}$. A probability density function for the time field peak at $\tau$ can be derived within a specific range [min, max]:

$$p(M = \tau) = \tau^{-\alpha}/c \tag{10}$$

where

$$C = \int_{\min}^{\max} M^{-\alpha}\, dM \tag{11}$$

is the power-law function integrated over [min, max]. Dividing the power-law function by $C$ ensures the area between [min, max] sums to 1, which makes it a probability distribution within the bounds. When $\alpha \neq 1$, the result of the integration is:

$$C = \frac{\max^{1-\alpha} - \min^{1-\alpha}}{1-\alpha} \tag{12}$$

When $\alpha = 1$, the result is:

$$C = \ln(\max) - \ln(\min) \tag{13}$$

As described above, we estimated the range through the Kolmogorov-Smirnov test prior to the hierarchical Bayesian model fitting following the procedure suggested by *Clauset et al., 2009*.

The value of $\alpha$ reflects the degree of compression in the distribution where a higher $\alpha$ value indicates that a bigger portion of time cells fire early in the delay. Crucially, when $\alpha$ approaches zero, *Equation 10* becomes $p(M = \tau) = \frac{1}{\max - \min}$, a uniform distribution between [min,max]. When sorting a group of evenly distributed time cells based on their temporal locations, one would expect the time fields to form a straight line. Another value of great theoretical interest for $\alpha$ is 1: where the time cells are distributed according to logarithmic compression. When sorting the log-compressed time cells on their temporal locations, one would expect to see more time cells early in the delay. The bottom panel of *Figure 2b* shows simulated time cells with these two $\alpha$ values.

## Log-normal Time Field Model

The Main Model assumes the shape of time fields to be symmetric but previous research (*Howard et al., 2014*) showed evidence for asymmetric time fields. Therefore, an alternative model assuming a log-normal shaped time field–an asymmetric receptive field–was applied to the data. Specifically, the Log-normal Time Field Model assumes that the probability of the observed spike at time t for trial i is

$$p_i(t) = a_1 \frac{1}{x\sigma\sqrt{2\pi}} \exp\left[-\frac{(\ln x - \mu)^2}{2\sigma^2}\right] + \frac{1-a_1}{l} \tag{14}$$

The Log-normal Tim Field Model is fitted in the same way as the Main Model except that the median of log-normal tuning curve, $\exp(\mu)$, is used as the time-field peak location for trial variability estimation at the cell level and temporal compression rate estimation at the population level. We used the standard deviation of the log-normal tuning curve for time field width, defined as

$$\sigma_n = \sqrt{[\exp(\sigma^2) - 1]\exp(2\mu + \sigma^2)} \tag{15}$$

## Alternative Trial Vary Model

The Main Model assumes the width of the time field stays consistent across trials, and the location of the time field shifts from trial to trial. Instead the Alternative Trial Vary Model assumes the time field location is fixed while the width varies across trials. Therefore the time field width for the specific trial $i$, $\sigma_{wi}$, is sampled from a normal distribution centered at $\sigma_W$:

$$\sigma_{wi} \sim \mathcal{N}(\sigma_W, \sigma_t^2) \tag{16}$$

## Exponential Compression Model and Weibull Compression Model

The Main Model assumes the compression at the population level is quantified with a power law distribution. Here, we tested two alternative common compression assumptions.

The Exponential Compression Model assumes that the time cell peaks M is distributed according to

$$p(M = \tau) = \beta \exp -\beta\tau \tag{17}$$

The Weibull Compression Model assumes that time field peaks M is distributed as

$$p(M = \tau) = \frac{k}{\lambda}\left(\frac{\tau}{\lambda}\right)^{k-1} \exp\left[-(\frac{\tau}{\lambda})^k\right] \tag{18}$$

The $k$ parameter controls the compression rate and results in a wide range of shapes that can be used to describe the asymmetry we clearly observed in time cell peak distributions. Specifically, when $k < 1$, more time cells would fire early; when $k = 1$, it becomes the exponential distribution; and when $k > 1$, it can approximate some heavy tail distributions such as ex-Gaussian or log-normal (*Logan, 1995*).

## Acknowledgements

The authors gratefully acknowledge support from ONR MURI N00014-16-1-2832, NIBIB R01EB022864, NIMH R01MH112169, NIMH R01MH095297, and NIMH R01MH132171. The authors gratefully acknowledge the contributions of Howard Eichenbaum to designing the behavioral task and data collection in this study.

## Additional information

### Funding

| Funder | Grant reference number | Author |
| --- | --- | --- |
| Multidisciplinary University Research Initiative | N00014-16-1-2832 | Rui Cao |

| Funder | Grant reference number | Author |
|---|---|---|
| National Institute of Biomedical Imaging and Bioengineering | R01EB022864 | Rui Cao |
| National Institute of Mental Health | R01MH112169 | Rui Cao |
| National Institute of Mental Health | R01MH095297 | Rui Cao |
| National Institute of Mental Health | R01MH132171 | John H Bladon |

The funders had no role in study design, data collection and interpretation, or the decision to submit the work for publication.

### Author contributions

Rui Cao, Formal analysis, Visualization, Methodology, Writing – original draft; John H Bladon, Data curation, Investigation, Writing – original draft; Stephen J Charczynski, Data curation, Software; Michael E Hasselmo, Conceptualization, Resources, Supervision, Funding acquisition; Marc W Howard, Conceptualization, Supervision, Funding acquisition, Project administration, Writing - review and editing

### Author ORCIDs

Rui Cao  http://orcid.org/0000-0003-0538-5336
John H Bladon  http://orcid.org/0000-0001-8993-9898
Marc W Howard  http://orcid.org/0000-0002-1478-1237

### Ethics

All procedures were conducted in accordance with the requirements set by the National Institutes of Health, and were approved by the Boston University Institutional Animal Care and Use Committee (BU IACUC protocol #16-021). Animals were given ad-libitum water and maintained at a minimum of 85% of their ad libitum feeding body weight during all behavioral training and testing. Surgeries were performed under isoflurane anesthesia, and analgesics were administered postoperatively.

### Decision letter and Author response

Decision letter https://doi.org/10.7554/eLife.75353.sa1
Author response https://doi.org/10.7554/eLife.75353.sa2

## Additional files

### Supplementary files
• Transparent reporting form

### Data availability

The data and code for all the analysis is available on Open Science Framework at https://osf.io/pqhjz/.

The following dataset was generated:

| Author(s) | Year | Dataset title | Dataset URL | Database and Identifier |
|---|---|---|---|---|
| Cao R, Bladon JH, Charczynski SJ, Hasselmo ME, Howard MW | 2021 | Weber-Fechner Time cells | https://osf.io/pqhjz/ | Open Science Framework, pqhjz |

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

## Appendix 1

### Supplementary Materials

### S1: Rats engaged stereotypical behavior with little variability after 2 s

In the Main Model, we found the temporal variability carried out by time cells linearly increases with delay. Given the well-documented interaction between timing and sensorimotor control (for review, see *Balasubramaniam et al., 2021*) , it is crucial to rule out the possibility that the widened time field for later time cells was simply due to some behavioral pattern that happens to be more variable towards the end of the delay. When we examined the average spatial location of each rat across the delay (*Figure 3—figure supplement 1a*), we saw little evidence suggesting the locations varied systematically across the delay, probably due to the absence of task demand for timing. In addition, we measured the variability of the stereotyped behavior engaged by animals below and found the variability was highest at the beginning, contrasting with the increased variability in the temporal information carried out by time cells.

We quantified two complementary measures of repetitive or stereotyped running behavior during the delay: spatial dispersion and head wobble distance. Spatial dispersion is the standard deviation of the Euclidean distance of a cluster of points in 2d space around a centroid. We measured the spatial dispersion of the rat's head at each second during the delay from the whole delay centroid to determine whether the rat behaved or positioned himself in a unique or stereotyped manner during any moment during the delay, which would amount to a dancing or counting behavior. Indeed, only in the first second during the delay did the mean dispersion exceed that across the remainder of the delay (*Figure 3—figure supplement 1b* top, under Friedman test, post-hoc ranksum second 1 was found different from all other seconds at p<.05). We then performed a complementary analysis on stereotyped running behaviors during the treadmill run. Some rats galloped on the treadmill, taking long strides and oscillating their bodies forward and backward, while others ran at a trot on the treadmill keeping their head more stationary, and yet others did a combination of the two. To quantify these two behaviors, we measured the total head wobble distance each rat covered at each second during the delay using unsmoothed position data. Indeed, rat 1 trotted during all delays, rats 2, and 4 performed a combination of the two, and rat 3 galloped almost entirely. Upon quantifying these behaviors during each second of the run, however, we found there is no significant difference in wobble across delay time except for in the first second (*Figure 3—figure supplement 1b* bottom, under Friedman test, post-hoc ranksum second 1 was found different from seconds 3–8 at p<.05).

### S2: Maximum Likelihood Fits

In this section, we report the results from the maximum likelihood fits for time cell selection (*Tiganj et al., 2018*; *Cruzado et al., 2020*) and compare them with fits from the Main Model. In *Figure 5—figure supplement 1a*, we compare the maximum likelihood estimated time field peaks as a function of the Main Model estimation. As shown in the figure, aside from a few outliers cluster toward the bottom, most of the cells were closely alongside the diagonal line, which suggests that the results from both methods are well aligned (r(129)=.92, p<.001 under Pearson correlation test). Similarly, when plotted on the log scale, the sorted time field peaks from Maximum likelihood fits tiled the delay evenly on the log scale (*Figure 5—figure supplement 1c*).

In *Figure 5—figure supplement 1b*, we plotted the maximum likelihood estimated time field width as a function of the estimation from the Main Model. Contrary to the time field peaks plot (*Figure 5—figure supplement 1a*), most of the cells here were above the diagonal line, which means the time field widths estimated under the maximum likelihood method tend to be larger than the ones from the Main Model. A paired t-test comparing the two confirmed that the difference was significant: t(131)=9.38, p<.001, with 95 percent confidence interval of [.34, .52]. The Bayesian paired t-test came to the same conclusion with . This result is expected as the maximum likelihood estimation does not separate the time field width from the across-trial variability. Additionally, in the Results section we reported a significant linear relationship between the estimated time field widths and time field peaks from the Main Model. However, when the same linear regression analysis was applied to the maximum likelihood estimated time field widths as a function of their time field peaks, both linear (.64±.06 p<.001) and quadratic term (-.06±.01, p<.001 ) were significant ($R^2$=.674, *Figure 5—figure supplement 1d*). Again it is likely because the maximum likelihood method included the trial-variability in the time field width estimation.

