## [Editor Report]

This is a rigorous evaluation of whether the compression of time cells in the hippocampus follows the Weber-Fechner Law, using a hierarchical Bayesian model that simultaneously accounts for the firing pattern at the trial, cell, and population levels. The two key results are that the time field width increases linearly with delay, even after taking into account the across trial response variability, and that the time cell population is distributed evenly on a logarithmic time scale.

---

## [Decision Letter]

**Decision letter after peer review:**

Thank you for submitting your article "Internally Generated Time in the Rodent Hippocampus is Logarithmically Compressed" for consideration by *eLife*. Your article has been reviewed by 3 peer reviewers, one of whom is a member of our Board of Reviewing Editors, and the evaluation has been overseen by Joshua Gold as the Senior Editor. The reviewers have opted to remain anonymous.

Recommendations for the authors:

The authors state that "The goal of the analyses in this paper is to rigorously test the hypothesis that time cells form a logarithmically-compressed representation of past time." While the authors made great strides and this rigorous analysis, unfortunately, no other hypothesis was really considered. It is common in timing tasks that subjects learn some rote behavioral sequence that happens to match the target interval. Therefore, correlates observed anywhere in the brain cannot uniquely be attributes to time, or that behavioral sequence. Moreover, it is not clear in the present report if the animals have actually learned the target duration and therefore have some representation of that duration. To pose the question in an actionable and concrete manner: what differs about the trials when time fields occur earlier than expected versus later? I strongly recommend that the authors classify each trial as an early firing versus late firing trial (or some other demarcation) and revisit the videos to assess whether animal behavior can explain any variability. My guess is that the behavior is more variable late into the treadmill running, and this could likely explain the variability of the firing fields.

Another major comment pertain to the nature of the statistical modeling. It is not totally clear what data is being used for the goodness-of-fit analysis, though it appears to be single trial firing rates. Are these smoothed? What is the bin size? These issues are important because a main focus of the manuscript is on trial-to-trial variability. The authors use their model to describe a conditional intensity function (equation 2), which is then tested directly with the spiking output. Typically, spikes are thought of as being probabilistically generated from an underlying intensity function and often Poisson link functions are used to translate between the world of stochastic events (spikes) and the world of the underlying, deterministic intensity functions (e.g. equation 2). How can the authors justify a direct estimation of the intensity function?

The authors test which coefficient best describes the power-law distribution of M in the hierarchical bayes model. No other distribution is considered, and then much of the paper is dedicated to a description of the implication of logarithmically compressed time field allocation. First, other distributions should be considered and compared (i.e. with AIC), and second, credible interval analysis should be used to motivate that the coefficient of the power law is not different from one (e.g. Keysers et al., 2020, https://dx.doi.org/10.1038%2Fs41593-020-0660-4).

Other models, and empirical observations, suggest skewed receptive fields that significantly differ from Gaussian – do the findings hold if the assumption of a Gaussian tuning is relaxed and other functions are considered (e.g. α function)?

The pairwise analysis for consistent shifts in the time field location is interesting though the effect seems modest. Two points, (1) since the authors are working within a Bayesian framework, confidence intervals and effect sizes can be given through an analysis of credible intervals (Keysers et al., 2020, https://dx.doi.org/10.1038%2Fs41593-020-0660-4), (2) Is it possible to leverage more than just a pairwise analysis since the authors conducted high density recordings. Several recent methods exist for such an analysis, such as Williams et al., 2020 https://doi.org/10.1016/j.neuron.2019.10.020, or Kawabata et al., 2020 https://doi.org/10.1152/jn.00200.2020. Similarly, a hidden markov model approach could be fruitful as well (Chen et al., 2012, https://dx.doi.org/10.1007%2Fs10827-012-0384-x).

In Figure 4c, the curved dashed line shows a sparsification of the putative temporal representation of long intervals. One issue is that, as the authors quantify, the time fields at the later delays are broader and show more variability. How does the compression differ with more relaxed inclusion criteria? This issue is important as the inclusion criteria (“a Gaussian-shaped time field, with a mean and a standard deviation, and a constant term”) select specifically for fields that show high trial-to-trial reliability (e.g. early fields). The authors must report how many total neurons per rat were recorded and what percentage was included/excluded.

Figure 5b – To my eye, it looks like there are a number of cells with low width throughout the interval hovering around the bottom of this graph and, at later peak times, across a gap from 0.25-0.5. I’m curious what the authors think about these cells. Are they merely outliers, or are there possibly subsets of cells that retain strong temporal fidelity even far into the interval? Is it possible to use unsupervised learning techniques like k-means clustering to find differences between these cells and the other ones which seem to follow the logarithmic law? Are there differences in the overall firing rates or any other properties? If there end up being different classes of cells, that would not invalidate the overall conclusion, but could be a novel discovery, And it could make the relationship with the other set of cells even stronger.

If I were of the opinion that a number of cells simply fired due to the onset of the event (as they would to an event boundary) and had different onset times as a result of the beginning of the interval due to the sudden change (Bright et al., 2020) but did not believe that variability increased thereafter, how would the authors address this? I realize this is tricky because it is not clear at what time point one may not consider a response to be directly related to some onset, but if the relationship in Figure 4b held when only investigating cells with peak firing times between two seconds and the end of the interval, that would address the concern.

In the regression analysis at the bottom of page 7, is the argument that the linear significance is relevant here? If so, the authors should make this more explicit, especially because the quadratic aspect looks so prominent.

“Section S1 of the supplementary materials provides an elementary explanation of why logarithmic compression leads to these properties.” It would be preferable to include a brief description of this logic in the main paper.

In discussing the Nielson Study, it seems highly relevant that the relationship between representational similarity and both space and time was logarithmic. It would also be nice to include the neurophysiological timing literature on the scalar property of timing. Recordings in the primate medial premotor areas during rhythmic timing have shown neural sequences linked with temporal processing (Crowe et al., 2014i; Merchant et al., 2015 i) and neural population codes that could underpin the scalar property of timing (Averbeck and Merchant, 2017; Gamez et al., 2019).

---

## [Author Response]

Recommendations for the authors:The authors state that “The goal of the analyses in this paper is to rigorously test the hypothesis that time cells form a logarithmically-compressed representation of past time.” While the authors made great strides and this rigorous analysis, unfortunately, no other hypothesis was really considered. It is common in timing tasks that subjects learn some rote behavioral sequence that happens to match the target interval. Therefore, correlates observed anywhere in the brain cannot uniquely be attributes to time, or that behavioral sequence. Moreover, it is not clear in the present report if the animals have actually learned the target duration and therefore have some representation of that duration. To pose the question in an actionable and concrete manner: what differs about the trials when time fields occur earlier than expected versus later? I strongly recommend that the authors classify each trial as an early firing versus late firing trial (or some other demarcation) and revisit the videos to assess whether animal behavior can explain any variability. My guess is that the behavior is more variable late into the treadmill running, and this could likely explain the variability of the firing fields.

The revision thoroughly explored the impact arkovioral variability on time cells, both within the delay and across trials and as predicted by the reviewers we observed a strong behavioral predictor of variability in time field location across trials. The original submission and the revision take the moment the animal crosses a laser beam on the treadmill as time zero for each trial. In the revision, we asked whether some variability in time field location across trials could be accounted for by instead using the moment the animal touches the treadmill (Figure 4b in the revision). For 3 out of the 4 animals, this variable accounted for trial-to-trial variability in some time cells, especially early in the delay. The section “Variability in the timing of events early in the trial accounts for some neural trial variability” (pp. 8-9) describes these results in more detail.

However, the revision also makes it clear that there is no evidence that behavioral variability across trials accounts for the evidence for logarithmically compressed time cells within a trial. The revision includes data about behavior as a function of time (e.g., Figure 3–supplement figure 1a). Animals moved from the front to the back of the treadmill over the first two seconds of the delay (accompanied by a lot of behavioral variability across trials) and were relatively stationary thereafter (see Supplementary S1). If behavioral variability could account for within trial effects (equations 3,4) we would expect time cells that peak before 2 s and after 2 s to have very different properties from the population as a whole. This was not the case (p.14, section “Time cells early and late in the delay show similar compression”). We thank the reviewers for raising the issue of behavioral variability. The additional analyses have led to a cleaner presentation (the revision de-emphasizes the importance of within-trial correlations) and a stronger empirical case that the logarithmic compression within-trial is not an artifact of behavior.

As a secondary question, it is true that in this study there is no behavioral demand for the animals to learn the delay interval and there is no evidence that the animals learned the delay. However, “hippocampal time cells” appear to be found in many experiments that do not explicitly require animals to learn a delay interval (e.g., Cruzado, et al., 2020; Taxidis, et al., 2020; Goh, 2021; Schonhaut, et al., 2022). If anything, the lack of an explicit timing demand would seem to reduce the incentive for animals to adopt a complex behavioral strategy as a proxy for time, thus reducing concerns that time cell phenomena observed in these experiments is merely a confound of a behavioral strategy.

Another major comment pertain to the nature of the statistical arkovng. It is not totally clear what data is being used for the goodness-of-fit analysis, though it appears to be single trial firing rates. Are these smoothed? What is the bin size? These issues are important because a main focus of the manuscript is on trial-to-trial variability. The authors use their model to describe a conditional intensity function (equation 2), which is then tested directly with the spiking output. Typically, spikes are thought of as being probabilistically generated from an underlying intensity function and often Poisson link functions are used to translate between the world of stochastic events (spikes) and the world of the underlying, deterministic intensity functions (e.g. equation 2). How can the authors justify a direct estimation of the intensity function?

The revision makes the method used in this paper more clear (section “Estimating time fields with hierarchical Bayesian method”, starting with line 130, p. 6). Briefly, the method does not estimate the intensity function, at the cost of being blind to overall firing rates. Instead the model assumes that a proportion of the observed spikes on each trial is independently sampled from the time field, specified by parameters for location and width. The goal of this method is to characterize the parameters of the time field at the trial level so those parameters can be fed to the next levels of the hierarchical Bayesian models. This is appropriate in that our hypothesis is specified in terms of the parameters of time fields (equations 3, 4). Although this method is limited in its utility, it is sufficient to ask this theoretically important question.

The authors test which coefficient best describes the power-law distribution of M in the hierarchical bayes model. No other distribution is considered, and then much of the paper is dedicated to a description of the implication of logarithmically compressed time field allocation. First, other distributions should be considered and compared (i.e. with AIC), and second, credible interval analysis should be used to motivate that the coefficient of the power law is not different from one (e.g. Keysers et al., 2020, https://dx.doi.org/10.1038%2Fs41593-020-0660-4).

In response to the first point, the revision considers several alternatives to the power law distribution of time field peaks (Table 1). The revision refers to the model from the initial submission as the “Main Model” and evaluates several alternative models. Two of these models, the Exponential Compression Model and the Weibull Compression Model, evaluate alternatives to the assumptions that time field peaks are distributed according to a power law. The exponential distribution and Weibull distribution are standard alternative distributions to test against power-law distribution (Clauset et al., 2009). The Main Model, which assumes a power-law distribution, outperforms both of these alternatives (Table 1). The observed spiking data is astronomically more likely if the values were generated from a power law distribution than either exponential or Weibull distributions. Detailed descriptions of the two alternative models can be found in section “Exponential Compression Model and Weibull Compression Model” (p. 28). Notably, even in the models where the model assumed that the distribution of time field peaks was not power law (equation 4), the parameters generated by these models still supported the other prediction of a logarithmically-compressed timeline (equation 3). As discussed in the revision (p. 12, “Alternative models …”) the width of receptive fields increased linearly with peak time when assuming both exponential and Weibull distributed peaks. We thank the reviewers for raising this issue; the inclusion of the alternative models allows the revision to make a much stronger empirical case for logarithmically-compressed time than did the original submission.

In response to the second point, the revision includes a 95% credible interval [.73, 1.11] for the exponent of the power law. Furthermore, about 54% of the posterior density is between the values of.9 and 1.1. Thank you for pointing out the Keysers et al., paper. That method is not applicable here as there is no prior in the analyses for the shape or the value of the exponent and the revision reports the posterior distribution in full. There is no null hypothesis in this context to generate the Bayes factor; performing additional statistical tests in Keysers et al., (2020) directly on the posterior distribution would violate the “independent sampling” assumption.

Other models, and empirical observations, suggest skewed receptive fields that significantly differ from Gaussian – do the findings hold if the assumption of a Gaussian tuning is relaxed and other functions are considered (e.g. α function)?

The revision makes clear that the answer to this question is “yes”. In the revision the Log-normal Time Field Model assumes skewed log-normal receptive fields (p. 27, line 878). As shown in Table 1, the Log-normal Time Field Model gives a somewhat inferior fit to the Main Model (which assumes Gaussian fields). The parameters of the Log-normal Time Field Model show similar evidence for both of the quantitative predictions of a logarithmic timeline (equations 3,4, see p. 12, line 331) as the Main Model.

The pairwise analysis for consistent shifts in the time field location is interesting though the effect seems modest. Two points, (1) since the authors are working within a Bayesian framework, confidence intervals and effect sizes can be given through an analysis of credible intervals (Keysers et al., 2020, https://dx.doi.org/10.1038%2Fs41593-020-0660-4), (2) Is it possible to leverage more than just a pairwise analysis since the authors conducted high density recordings. Several recent methods exist for such an analysis, such as Williams et al., 2020 https://doi.org/10.1016/j.neuron.2019.10.020, or Kawabata et al., 2020 https://doi.org/10.1152/jn.00200.2020. Similarly, a hidden arkov model approach could be fruitful as well (Chen et al., 2012, https://dx.doi.org/10.1007%2Fs10827-012-0384-x).

The revision does not include any data on pairwise correlations among time cells. After carefully considering the results of additional analyses showing that time field peaks can vary across trials in tandem with behavioral variability (in response to point 1) we decided that there is no reason to conclude that pairwise correlations are evidence for an “internal timeline”. Insofar as time cells are correlated with behavioral variables it is not surprising they are correlated with one another. It would be meaningful to say the timeline is “internal” if one had measured behavior more extensively (e.g., with deeplabcut) and still found correlations between time cell peaks after controlling for behavior. However, this question is not central to the question of whether the timeline is logarithmic within trial.

In Figure 4c, the curved dashed line shows a sparsification of the putative temporal representation of long intervals. One issue is that, as the authors quantify, the time fields at the later delays are broader and show more variability. How does the compression differ with more relaxed inclusion criteria? This issue is important as the inclusion criteria ("a Gaussian-shaped time field, with a mean and a standard deviation, and a constant term") select specifically for fields that show high trial-to-trial reliability (e.g. early fields). The authors must report how many total neurons per rat were recorded and what percentage was included/excluded.

The revision reports the total number of units recorded and included/excluded per rat on page 6. Overall, the fraction of time cells, 159/625, was in the same ballpark as previous time cell papers (e.g., Salz et al., 2016). In addition, only a small portion of units, 6/159, was impacted by relaxing the criterion for including time cells (p. 25, line 784). Notably, none of those cells had a peak after 4 s. Thus the revision makes a much stronger case that the conclusions about the number density of units were not an artifact of a specific criterion for including time cells in the hierarchical Bayesian analyses. We thank the reviewers for suggesting these changes, which have resulted in a much stronger empirical case.

Figure 5b – To my eye, it looks like there are a number of cells with low width throughout the interval hovering around the bottom of this graph and, at later peak times, across a gap from 0.25-0.5. I'm curious what the authors think about these cells. Are they merely outliers, or are there possibly subsets of cells that retain strong temporal fidelity even far into the interval? Is it possible to use unsupervised learning techniques like k-means clustering to find differences between these cells and the other ones which seem to follow the logarithmic law? Are there differences in the overall firing rates or any other properties? If there end up being different classes of cells, that would not invalidate the overall conclusion, but could be a novel discovery, And it could make the relationship with the other set of cells even stronger.

The revision makes clear that the apparent cluster of cells with narrow width and broad time field peak in the previous submission was largely due to units with low spike count within the time field—several of the units have less than two spikes attributable to the time field per trial. The revision uses shading in the scatterplot to indicate the number of spikes under the time field (see Figure 5b). Note that the number of spikes per trial attributable to the time field is distinct from the number of spikes per trial, which was a criterion for inclusion in the hierarchical analyses.

If I were of the opinion that a number of cells simply fired due to the onset of the event (as they would to an event boundary) and had different onset times as a result of the beginning of the interval due to the sudden change (Bright et al., 2020) but did not believe that variability increased thereafter, how would the authors address this? I realize this is tricky because it is not clear at what time point one may not consider a response to be directly related to some onset, but if the relationship in Figure 4b held when only investigating cells with peak firing times between two seconds and the end of the interval, that would address the concern.

The revision explicitly shows that the relationship in figure 3b holds “when only investigating cells with peak firing times between two seconds and the end of the interval”. Fortuitously, the choice suggested by the reviewers also does a good job of separating periods with high behavioral variability and low behavioral variability (Figure 3–supplement figure 1, response to point 1 above). The section of the revision entitled “Time cells early and late in the delay show similar compression” (p. 14, line 355) provides details. Briefly, there is an increase in time field width as a function of peak time (the relationship in Figure 5b) for both groups. Dividing the population into two groups we lose statistical power and can’t make statements with the same precision (e.g., it’s not possible to argue that the peaks are distributed with power law with exponent near 1, but simply to argue that the peaks are not uniformly distributed). More broadly, there is no evidence that the groups differ from one another despite a dramatic difference in behavioral variability between those time periods. Thank you to the reviewers for raising this issue, which has enabled the revision to rule out an alternative interpretation of the results.

In the regression analysis at the bottom of page 7, is the argument that the linear significance is relevant here? If so, the authors should make this more explicit, especially because the quadratic aspect looks so prominent.

The revision is careful not to make any claim that the linear term is of special importance for across-trial variability. The text (section “Across-trial variability changes over the delay interval”, p. 9) and the caption of Figure 4 make clear that the regression describing across trial variability as a function of peak time produced reliable intercept, linear and quadratic terms. Note that the question of across-trial variability is orthogonal to the questions about within-trial temporal receptive fields which are central to the hypothesis of the paper. By contrast, the same Bayesian regression analysis for the within-trial fields preferred the linear-term-only model over alternative models including one with both the linear term and the quadratic term (detailed analysis on p. 11, line 274).

"Section S1 of the supplementary materials provides an elementary explanation of why logarithmic compression leads to these properties." It would be preferable to include a brief description of this logic in the main paper.

The revision integrates the elementary explanation of how logarithmic compression leads to two particular quantitative relationships (equations 3,4) into the introduction of the revision (“Quantifying logarithmic compression”, starting on p. 2, line 44). This was an excellent suggestion as it focuses the reader on two well-specified quantitative questions that frame the empirical results.

In discussing the Nielson Study, it seems highly relevant that the relationship between representational similarity and both space and time was logarithmic. It would also be nice to include the neurophysiological timing literature on the scalar property of timing. Recordings in the primate medial premotor areas during rhythmic timing have shown neural sequences linked with temporal processing (Crowe et al., 2014i; Merchant et al., 2015 i) and neural population codes that could underpin the scalar property of timing (Averbeck and Merchant, 2017; Gamez et al., 2019).

The discussion of the revision makes both of these points. The discussion points out that the Nielsen et al., (2015) paper observed logarithmic compression in both time and space (p. 21, line 641). The discussion also discusses the relationship between the results observed here and evidence for scale-invariant representations of time and sequence (“Implications of logarithmically-compressed time…”, p.18). Briefly, logarithmically-compressed time cells are scale-covariant; rescaling time in an experiment would result in translation of the activity across the set of neurons. Figure 6, which is new to the revision, clarifies this idea. Scalar timing models—more generally scale-invariant properties—can be computed from a scale-covariant representation. The confluence of these kind of results across many different brain regions suggests a design principle for the brain’s estimates of time.